# The induction of natural competence adapts staphylococcal metabolism to infection

Mar Cordero[1,10], Julia García-Fernández [1,10], Ivan C. Acosta [1], Ana Yepes[2,3], Jose Avendano-Ortiz[4], Clivia Lisowski[3], Babett Oesterreicht[2,3], Knut Ohlsen[2,3], Eduardo Lopez-Collazo[4,5], Konrad U. Förstner [2,3,6,7], Ana Eulalio [3,8,9] & Daniel Lopez [1,2,3✉]

A central question concerning natural competence is why orthologs of competence genes are conserved in non-competent bacterial species, suggesting they have a role other than in transformation. Here we show that competence induction in the human pathogen *Staphylococcus aureus* occurs in response to ROS and host defenses that compromise bacterial respiration during infection. Bacteria cope with reduced respiration by obtaining energy through fermentation instead. Since fermentation is energetically less efficient than respiration, the energy supply must be assured by increasing the glycolytic flux. The induction of natural competence increases the rate of glycolysis in bacteria that are unable to respire via upregulation of DNA- and glucose-uptake systems. A competent-defective mutant showed no such increase in glycolysis, which negatively affects its survival in both mouse and *Galleria* infection models. Natural competence foster genetic variability and provides *S. aureus* with additional nutritional and metabolic possibilities, allowing it to proliferate during infection.

[1] National Centre for Biotechnology, Spanish National Research Council (CNB-CSIC), 28049 Madrid, Spain. [2] Research Centre for Infectious Diseases (ZINF), University of Würzburg, 97080 Würzburg, Germany. [3] Institute for Molecular Infection Biology (IMIB), University of Würzburg, 97080 Würzburg, Germany. [4] The Innate Immune Response and Tumor Immunology Group, IdiPaz La Paz University Hospital, 28046 Madrid, Spain. [5] CIBER of Respiratory Diseases (CIBERES), Madrid, Spain. [6] Information Centre for Life Science (ZBMED), 50931 Cologne, Germany. [7] TH Köln – University of Applied Sciences, 50578 Cologne, Germany. [8] Center for Neuroscience and Cell Biology (CNC), University of Coimbra, 3004-504 Coimbra, Portugal. [9] Institute of Biomedicine (iBiMED), Department of Medical Sciences, University of Aveiro, 3810-193 Aveiro, Portugal. [10]These authors contributed equally: Mar Cordero, Julia García-Fernández. ✉email: dlopez@cnb.csic.es

Living organisms must be able to adapt their physiology to changing conditions. When infecting a host, pathogenic bacteria must cope with a hostile environment, mostly caused by the accumulation of immune cells to the infection site, which are programmed to attack and eliminate the invading microbes. The success of a pathogen partly resides in its capacity to survive and fight against these host defenses, which enable bacteria to colonize diverse organs and cause different types of infections. One such microbe is the human pathogen *Staphylococcus aureus*, an important hospital-associated bacterial pathogen. It can innocuously colonize the nasopharynx and skin, becoming pathogenic when either the latter or a mucous membrane barrier is compromised and the pathogen is internalized into the body[1]. It can cause a variety of severe soft tissue and bone infections, and occasionally life-threatening necrotizing fasciitis or pneumonia[2]. Furthermore, the widespread use of antibiotics in recent decades has allowed *S. aureus* to gradually acquire antimicrobial resistance determinants via horizontal gene transfer (HGT); multi-drug resistant strains of *S. aureus* are now endemic in hospitals worldwide (HA-MRSA)[3] and also are becoming prevalent in communities of individuals without previous healthcare contact (CA-MRSA). The high incidence of infections caused by *S. aureus*, and the need for new ways to prevent and treat them, requires a deeper understanding of the pathogenesis of staphylococcal disease.

The success of *S. aureus* as a pathogen lies in its acquisition of antibiotic resistance but also in its ability to metabolically adapt to different sites in the human body to overcome the host innate immune response[4]. During infection, *S. aureus* proliferation is driven by aerobic or anaerobic respiration and fermentation[4]. The latter is a key bioenergetic pathway during staphylococcal infections[5,6], since many host tissue conditions demand non-respiratory growth by the pathogen. When the immune system detects the presence of a pathogen, the influx of neutrophils to the infection site is accompanied by the production of reactive oxygen species (ROS)[7,8] that inhibit bacterial growth. Indeed, immune cell-produced ROS cause important damage to bacterial DNA, lipids, and proteins, and seriously impede the biosynthesis of some enzymes with metal- and thiol-containing groups that participate in the TCA cycle and electron transport chain[9,10]. *S. aureus* respiration cannot, therefore, proceed as normal. The accumulation of activated neutrophils also leads to an increase in host cell oxygen consumption, driving down the availability of oxygen to the pathogen[11], further interfering with its respiration. The sequestration of iron (an essential cofactor of respiratory proteins) by host proteins also limits bacterial respiration, providing an innate protective mechanism against invading pathogens[12,13].

Pathogens reconfigure both metabolism and virulence to avoid clearance by the immune system and counteract the antibacterial actions of the host. *S. aureus* can counteract the inhibition of respiration caused by the host immune system by redirecting its metabolism towards fermentation. One of the most dramatic examples of this is seen in the appearance of small-colony variants (SCV), respiratory-deficient mutants that are not susceptible to such respiratory inhibition, allowing them to cause chronic and perhaps eventually life-threatening infections[14,15]. When not performing respiration, *S. aureus* uses fermentation to produce ATP[5,6]. However, fermentation is energetically less efficient than respiration, which reduces the energy output of fermenting *S. aureus* cells, reducing their growth rate. Indeed, under non-respiring growth conditions, *S. aureus* must consume far more glucose to reach the respiration-equivalent biomass[16]. To compensate, *S. aureus* increases the glycolytic flux during fermentation[16]. As glucose availability is limited in non-diabetic patients[17–22], *S. aureus* may activate specific cellular programs to facilitate a more efficient uptake of glucose or alternative catabolizable metabolites. The mechanism by which *S. aureus* enhances the glycolytic flux and therefore, provide to non-respiring *S. aureus* cells the required growth rate to thrive during infections is, however, unclear.

Natural competence allows bacteria to take up extracellular DNA fragments as a nutritional resource[23,24], and as a source of nucleotides for DNA synthesis[25] or homologous recombination[26,27]. The latter can lead to natural transformation and the appearance of antibiotic-resistant bacteria[28]. The precise role of natural competence in bacteria is still debated, partly because a practically full repertoire of competence gene orthologs (*com* genes) are conserved in the genomes of *S. aureus* and other non-competent species such as *Listeria monocytogenes* and *Lactococcus lactis*[29,30]. The role of these genes in these latter bacteria is unknown, suggesting that the competence machinery is involved in something other than natural competence in these species. Moreover, the induction of competence genes is rare in non-competent bacterial species, partly attributable to the undetectable expression of the ComK regulator, which triggers the expression of the *com* genes[31–35]. In *S. aureus*, *comK* expression is undetectable under standard laboratory growth conditions. However, strains engineered to overexpress *comK* are known to induce the genes required for natural competence[36].

Here we describe the role of ComK in increasing the rate of glycolysis in non-respiring *S. aureus*. Its induction enables *S. aureus* to obtain the energy required to grow during infection, in conditions in which bacterial respiration is inhibited, while promoting DNA uptake from the environment. We show that *comK* is strongly induced during infection and in response to ROS. ComK upregulates the genes that code for the glucose- and DNA-uptake transport machineries, providing additional nutrients to enhance the fermentation capabilities of bacteria that are unable to respire and a nucleotide source for repair of DNA damage caused by ROS. As expected, ROS were found to cause serious damage to respiration in a Δ*comK*, competence-deficient mutant, which led to the accumulation of intracellular ROS and DNA mutations in its chromosome. Its survival was therefore significantly reduced and it was strongly impaired in terms of causing infection in vertebrate and invertebrate infection models. Our work demonstrates that natural competence in staphylococci goes beyond fostering genetic variability, specifically to provide bacteria with additional nutritional and metabolic possibilities thus enabling them to proliferate during infection.

## Results

**Staphylococcus aureus induces comK during infection.** Several laboratories have reported natural competence in *S. aureus* cultures at low frequencies of DNA transfer[31,33], a phenomenon attributed to the very low basal expression levels of the ComK competence regulator (Supplementary Fig. 1A and B). ComK triggers natural transformation in *Bacillus subtilis*[37] and is a widely conserved regulator in other, non-competent firmicutes, suggesting that it plays an unknown role other than natural competence. The induction of DNA uptake in *S. aureus* requires *comK* overexpression to increase ComK levels[31–35]. We detected low basal *comK* expression under standard laboratory growth conditions (TSB medium incubated at 37 °C with 200 rpm agitation sampled at late-exponential or stationary phase) using qRT-PCR (Fig. 1Ai) or immunodetection assays (Supplementary Fig. 1B). In agreement with this, in a competence assay[31] with plasmid or chromosomal DNA carrying the *erm* and *lac* genes (which confer erythromycin resistance and β-galactosidase activity upon uptake, respectively), no *erm*+ *lac*+ colonies were detected (Fig. 1Aii and Supplementary Table 1). Varying the

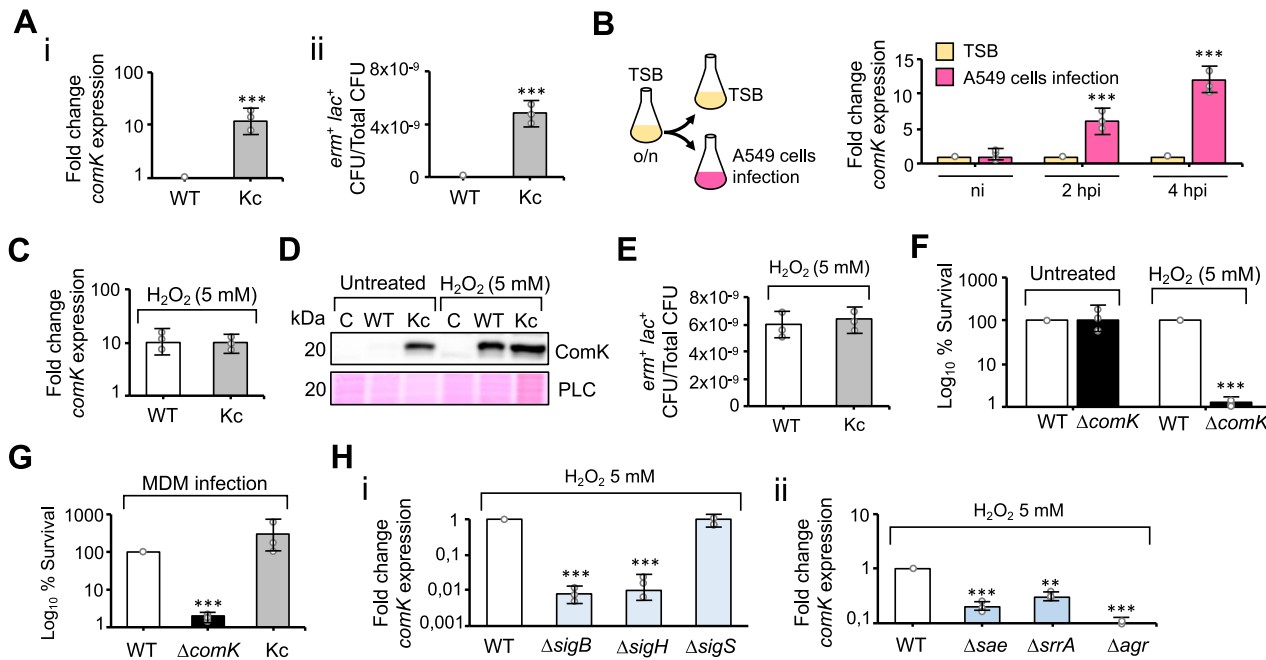

**Fig. 1 comK is induced in response to oxidative stress. A** (i) qRT-PCR analysis of *comK* expression in *S. aureus* WT (wild type) and Kc, a strain that constitutively expresses *comK* (TSB medium 37 °C, agitation 200 rpm). (ii) DNA uptake efficiency of different *S. aureus* strains. Data represent the number of CFU per ml of $erm^+$ $lac^+$ colonies in relation to the total CFU per ml (CFU/ml). Significance was measured by two-tailed Student *t* test; ***$p < 0.001$. Data are shown as mean ± SD of three independent experiments ($n = 3$). **B** qRT-PCR analysis of *comK* expression during *S. aureus* infection of A549 human-lung epithelial cells (MOI 100); analysis at 2 and 4 h post-infection (h.p.i.). The inoculum was obtained from a TSB culture. As a control, fresh TSB medium was inoculated with a comparable number of *S. aureus* cells. Statistical analysis, two-tailed Student *t* test; ***$p < 0.001$. Data are shown as mean ± SD of three independent experiments ($n = 3$). **C** qRT-PCR analysis of *comK* expression in WT and Kc strains in the presence of $H_2O_2$ (5 mM) Data are shown as mean ± SD of three independent experiments ($n = 3$). **D** Immunodetection of ComK protein levels in WT and Kc strains in $H_2O_2$-treated and untreated cultures. Ponceau membrane staining is shown as a protein loading control (PLC). These results are representative of the results obtained in three independent experiments. **E** DNA uptake efficiency of WT and Kc strains in TSB cultures supplemented with $H_2O_2$ (5 mM). Data are shown as mean ± SD of three independent experiments ($n = 3$). **F** Survival (%) of different *S. aureus* strains in untreated TSB cultures or TSB cultures with $H_2O_2$ (5 mM). The survival rate was determined when cultures reached the mid-exponential phase. Statistical analysis, two-tailed Student *t* test; ***$p < 0.001$. Data are shown as mean ± SD of three independent experiments ($n = 3$). **G** *S. aureus* survival (%) in infected human monocyte-derived macrophages (MDM) (MOI 10) at 2 h.p.i. Differences were examined by one-sided ANOVA with Tukey's test for multiple comparisons; ***$p < 0.001$. Data are shown as mean ± SD of three independent experiments ($n = 3$). **H** qRT-PCR analyses of *comK* expression in different *S. aureus* mutants in TSB cultures supplemented with $H_2O_2$ (5 mM). Differences were detected by one-sided ANOVA with Tukey's test for multiple comparisons; ***$p < 0.01$, ***$p < 0.001$. Data are shown as mean ± SD of three independent experiments ($n = 3$). Source data are provided as a Source Data file.

incubation time, temperature and degree of aeration failed to induce *comK* expression, nor were $erm^+$ $lac^+$ colonies detected in competence assay (Supplementary Table 1). In contrast, a strain constitutively expressing *comK* (Kc) (Fig. 1Ai) showed $erm^+$ $lac^+$ colonies in competence assays when using plasmid DNA ($10^{-8}$–$10^{-9}$ positive CFU/total CFU) (Fig. 1Aii).

Experiments were next performed to determine the growth conditions that upregulate *comK* expression. Since *S. aureus* can colonize human lungs and cause pneumonia[38,39], among other infections, *comK* expression was evaluated during the infection of human alveolar epithelial cells (A549 cells) (Fig. 1B). The strong induction of *comK* expression was detected under these conditions, suggesting that infection-related cues trigger *comK* expression. To better understand the induction of *comK* expression during this in vitro infection, *S. aureus* laboratory cultures were grown in the presence of different infection-related cues, and *comK* expression determined. Cues associated with oxidative damage, such as exposure to $H_2O_2$ (5 mM), strongly induced *comK* expression (Fig. 1C, D and Supplementary Table 1). Under oxidative damage-inducing conditions (e.g., exposure to $H_2O_2$, acidic pH or mupirocin treatment), DNA uptake was increased to levels comparable to those observed for the Kc strain (Fig. 1E). A single dose of non-oxidant stressors did not induce *comK* or increase DNA uptake capacity

(Supplementary Table 1) under the growth conditions used. We generated a catalase-deficient mutant; catalases are enzymes involved in degradation of $H_2O_2$, which allow bacteria to better resist the oxidative stress of the immune response during infection[40]. The catalase-deficient mutant was highly sensitive to the presence of ROS and showed a constitutive induction of *comK* in TSB cultures compared to the basal *comK* expression in WT cultures (Supplementary Fig. 2). This result pointed to the direct role of ROS to induce *comK* and the competence response in *S. aureus*.

To determine the importance of *comK* induction in response to oxidative stress, a Δ*comK* mutant was generated and grown in the presence of $H_2O_2$ (5 mM). Under oxidative stress, this mutant returned a much lower CFU count than either wild type (WT) or Kc cultures (Fig. 1F and Supplementary Fig. 3A). No phenotypic differences were detected in untreated cultures (Supplementary Fig. 3B and C). In other *S. aureus* genetic backgrounds (the USA300 and 8325-4 strains), the Δ*comK* mutant also returned a much lower CFU count than the WT and Kc strains under oxidative stress conditions (Supplementary Fig. 4A). Overall, these results suggest that the Δ*comK* mutant is more sensitive to oxidative stress than the WT. The $H_2O_2$ treatment (5 mM) also caused a significant growth defect in the *B. subtilis* Δ*comK* mutant compared to the WT strain (168 strain) (Supplementary Fig. 4B),

suggesting that *comK* protects against oxidative stress in this bacterium as well as in *S. aureus*. ROS, including $H_2O_2$, are central players in the antimicrobial response of macrophages and other innate immune cells in bacterial infections[41,42] and ROS are known to modulate the competence response in close-related pathogens[43]. Thus, monocyte-derived macrophages (MDM) were infected with the different *S. aureus* strains. The infection of macrophages by the Δ*comK* mutant was strongly impaired compared to the WT and Kc strain (Fig. 1G).

**$H_2O_2$ induces *comK* expression via activation of *agrCA*, *saeRS*, and *srrAB*.** As ROS are known to affect the signaling cascade controlling natural competence in related pathogens[43,44], we aimed to understand the mechanism of how $H_2O_2$ induces *comK* expression during infection. We searched for the regulators that trigger *comK* expression in response to oxidative stress. In *B. subtilis*, ComK regulation is mediated by the adaptor protein MecA. The proteolytic complex of ClpP-C degrades ComK by directly binding to the adapter protein MecA[45,46]. A bacterial two-hybrid assay was used to identify the interactions between the MecA-homolog (SA0857 locus) and ComK. ComK–ClpP interaction was detected but no SA0857–ComK interaction was observed (Supplementary Fig. 5A and B). The immunodetection of ComK in a Δ*sa0857* mutant and the WT strain revealed no alteration in ComK levels in the former (Supplementary Fig. 5C). Thus, the SA0857-homolog protein of *S. aureus* does not seem to play an important role in ComK regulation under the growth conditions used.

*S. aureus* develops natural competence with the induction of the alternative sigma factor SigH, as SigH overexpression induces the staphylococcal *comG* and *comE* operons[31]. It is known that a simultaneous expression of *sigH* and *comK* is necessary to induce the genes required for staphylococcal transformation[36]. Thus, we evaluated whether *sigH*, as well as other alternative sigma factors, are involved in the induction of *comK* by $H_2O_2$ during infection. *S. aureus* possesses three alternative sigma factors: (i) SigB controls the response to heat, oxidative and antibiotic stresses[47–49], (ii) SigH is involved in the transcription of competence genes[36] and prophage integration/excision[50] and (iii) an extracytoplasmic function (ECF) SigS that responds to starvation and cell wall damage[51,52]. We generated knock-out mutants of these alternative sigma factors (Δ*sigS*, Δ*sigH*, or Δ*sigB*) and tested the induction of *comK* expression in $H_2O_2$-supplemented cultures (Fig. 1Hi and Supplementary Fig. 6A and B). A significant reduction in *comK* induction was detected in mutants of *sigB* and *sigH* (also in the mutant of the *sigB*-regulator *rswB*), whereas no differences were detected in the mutant of *sigS*. Thus, a SigB-mediated oxidative stress response may induce *comK* expression in *S. aureus*, in addition to the induction by SigH.

To gain more insight about the mechanism of *comK* induction in response to oxidative stress, a collection of knock-out mutants for regulators known to respond to $H_2O_2$ and related infection signals (*rswB*, *codY*, *ccpA*, *ccpE*, *rot*, *sarA*, *kdp*, *phoR*, *agrCA*, *srrAB*, *arl* and *saeRS*) was then tested to determine whether any showed reduced induction of *comK* expression in response to $H_2O_2$ (Supplementary Fig. 6A). Importantly, the presence of $H_2O_2$ did not induce *comK* expression in strains lacking the respiratory response two-component systems *srrAB*, *saeRS*, or *agrCA*, which trigger the expression of hemolytic toxins and other virulence factors[53] (Fig. 1Hii). The AgrA and SrrB response regulators sense $H_2O_2$ via the formation of an intramolecular disulfide bond (AgrA) or via redox-sensitive cysteines that respond to the status of the respiration quinone pool (SrrB)[54,55], whereas $H_2O_2$ induces the expression of the *saeRS* operon[56]. SrrAB and SaeRS are both

stimulated by defective respiration[57,58]. Thus, it is possible that a defective respiration induces *comK* expression via *saeRS*, *agrCA* or *srrAB* upregulation, suggesting that *comK* may be an important component of *S. aureus* virulence to respond to the oxidative damage caused by the host immune system.

**Inhibition of respiration induces *comK* expression in *S. aureus*.** To investigate the role of ComK in *S. aureus* pathogenesis, a *S. aureus* strain was generated harboring a chromosomally integrated $P_{comG}$-*yfp* transcriptional fusion to report on ComK activity. The *comG* operon encodes the proteins responsible for the assembly of the DNA transformation pillus[59]. In *S. aureus*, *comG* expression depends on ComK (Fig. 2A, B)[31,36] thus, the Kc strain showed stronger expression of $P_{comG}$-*yfp* than did the WT (Supplementary Fig. 7A), which showed no induction of the reporter in TSB medium. However, after a random mutagenesis experiment using N-methyl N-nitro N-nitrosoguanidine (NG), in which ~36,000 colonies were screened, a strain (NG8 strain) was isolated that showed a strong and bimodal expression of the reporter[60] (Fig. 2A, B and Supplementary Fig. 7A) as well as high endogenous *comK* expression levels (Fig. 2C). This NG8 strain showed a growth yield lower than that of the WT in TSB medium (at 37 °C with agitation at 200 rpm) (Fig. 2D). Genome sequencing of the NG8 strain revealed a total of 112 single nucleotide polymorphisms (SNPs) (Supplementary Table 2). 49 SNPs of the total caused important changes in annotated ORFs (2 nonsense mutations and 47 missense mutations) (Fig. 2E). The remaining SNPs were located in intergenic regions, causing silent mutations (i.e., a codon change without a subsequent change in the amino acid) or hypothetical low-impact mutations (i.e., a change to an amino acid of similar physicochemical properties) (Supplementary Fig. 7B). A large fraction of SNPs (26 SNPs, corresponding to 30% of the total SNPs) affected genes involved in carbohydrate metabolism and respiration (e.g., *cydA*, *cydB*, *ldh*, *pycA*, *pfkB*, or *glpD*).

The many SNPs associated with a potentially defective respiration led us to hypothesize that respiration inhibition might compromise energy production in the NG8 strain, and might be associated with the greater *comK* expression detected in this strain. Bacteria that are unable to respire produce energy by fermentation instead. However, fermentation is energetically less efficient than respiration, which may explain the lower growth yield of NG8 strain in TSB medium. Thus, the acidic end products of fermentation, such as acetate and lactate, were quantified in culture supernatants at different time points during the growth cycle (exponential, post-exponential, stationary and late stationary phases) and represented as a function of culture $OD_{600}$ (Fig. 2F). Fermentation end products were found in significantly higher concentrations for the NG8 strain than for the WT. In agreement with the hypothesis that NG8 strain shows a respiration-defective phenotype, the oxygen consumption rate, measured as an indicator of aerobic respiration, was lower in the NG8 strain than in the WT (Fig. 2G). Taken together, these results suggest that the NG8 strain, which shows *comK* expression induced, relies more on fermentation to obtain energy, as it is unable to respire in our growth assay.

To test whether non-respiring metabolism is associated with *comK* expression in other *S. aureus* strains, a number of hospital SCVs isolated from chronic infections were examined. A non-respiring metabolism is a hallmark of SCVs, which are typically defective in the electron transport chain or TCA cycle, which provides a slow growth that is beneficial to the fitness of these strains compared to WT strains during persistent infections and antibiotic therapy[14,15]. We tested a number of heme-defective SCV isolates associated with chronic infections and showed a

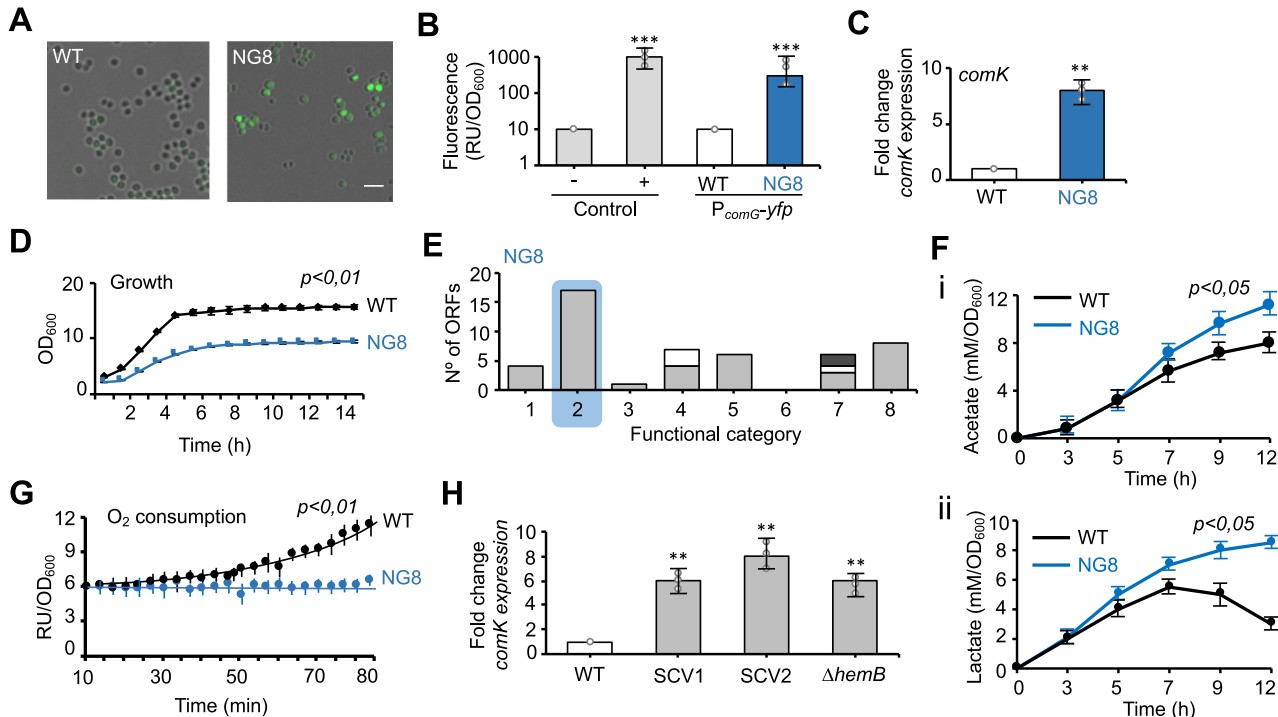

**Fig. 2 *comK* induction occurs in fermenting *S. aureus* strains. A** Fluorescence microscopy analysis of P*comG*-*yfp* expression in the WT and the nitrosoguanidine-treated strain NG8. The NG8 strain showed a heterogeneously distributed fluorescence signal in a subpopulation of cells. Scale bar = 10 μm. **B** Quantitative determination of the fluorescence signal in P*comG*-*yfp*-expressing WT and NG8 strains. The negative and positive controls were *B. subtilis* 168 WT and the P*comG*-*yfp* labeled strain, respectively. Significance was measured by two-tailed Student *t* test; ***$p < 0.001$. Data are shown as mean ± SD of three independent experiments ($n = 3$). **C** qRT-PCR analysis of *comK* expression in untreated TSB cultures. Statistical analysis, two-tailed Student *t* test; **$p < 0.01$. Data are shown as mean ± SD of three independent experiments ($n = 3$). **D** Growth curves of WT and NG8 strains in untreated TSB cultures. Two-tailed Student *t* test. Data are shown as mean ± SD of three independent experiments ($n = 3$). **E** Classification of ORFs that harbor SNPs in the strain NG8 using TIGRfam, SEED, and Gene Ontology functional categories. Represented ORFs harboring nonsense mutations and missense mutations with moderate effect. For each category, columns represent the number of regulated genes. Group 1—protein metabolism; group 2—carbohydrate metabolism and respiration; group 3—amino acid metabolism; group 4—stress response genes; group 5—virulence; group 6—iron acquisition; group 7—DNA metabolism (gray), cell division (white); group 8—other genes. Group 2 genes were more strongly represented in the NG8 strain (highlighted in blue). **F** Determination of acetate (i) and lactate (ii) levels in culture supernatants of the WT and NG8 strains. To measure acetate production, 10 ml of TSB cultures were grown in 100 ml Erlenmeyer flasks. To measure lactate production, 20 ml of cultures were grown in 100 ml Erlenmeyer flasks. Concentration is represented in relation to culture OD_{600}. Statistical analysis, two-tailed Student *t* test. Data are shown as mean ± SD of three independent experiments ($n = 3$). **G** Determination of oxygen consumption rate in WT and NG8 strains over time. Oxygen quenches the fluorescence signal of the dye (ab197243) added to the culture, thus oxygen consumption is directly related to fluorescence. Oxygen consumption rates are represented as relative fluorescence units in relation to cultures at OD_{600}. Cultures were grown in TSB medium for 24 h at 37 °C with agitation (200 rpm). Two-tailed Student *t* test. Data are shown as mean ± SD of three independent experiments ($n = 3$). **H** qRT-PCR analysis of *comK* expression in TSB cultures of small-colony variants (SCV). SCV1 and SCV2 are clinical isolates from the Hospital Ramón y Cajal (Madrid, Spain). Δ*hemB* is a laboratory mutant. Comparisons were made using one-sided ANOVA with Tukey's test for multiple comparisons; **$p < 0.01$. Data are shown as mean ± SD of three independent experiments ($n = 3$). Source data are provided as a Source Data file.

strong induction of *comK* expression in all strains tested (Fig. 2H). In addition, a Δ*hemB* mutant that reproduced the non-respiring SCV phenotype also showed a strong induction of *comK* expression, indicating that *comK* induction is linked to the inhibition of respiration in *S. aureus* strains.

**Induction of *comK* enhances the glycolytic capacity of *S. aureus* cells unable to respire.** The link between *comK* expression and the inhibition of respiration in *S. aureus* was next pursued. For this, global gene-expression profiling was undertaken by RNA-seq to identify differentially regulated genes in strains with different *comK* expression levels. Specifically, the expression profile of WT was compared to that of the Kc strain in untreated cultures (TSB, 37 °C, 200 rpm), and to those of the WT and Δ*comK* mutant in TSB cultures under oxidative stress. A total of 201 ComK upregulated and 118 downregulated genes were identified (Fig. 3A,

Supplementary Fig. 7C, Supplementary Table 3 and Supplementary Data 1). Genes involved in carbohydrate metabolism were detected among those upregulated (Fig. 3A–C and Supplementary Fig. 8A), particularly those associated with glycolysis (*pgi, fba, tpiA, glpD, pgk, pgm, eno, pyk, pdh*) and the phosphotransferase transport system, which brings sugars into the cytoplasm to fuel glycolysis. The induction of *cidC* and *alsSD* was detected, genes involved in the carbon-overflow pathways associated with fermentative metabolism[4,61]. We also detected the induction of genes associated with the pentose phosphate pathway (*gntKR, tkt, rpe*) and with purine and pyrimidine metabolism (*purF, purN, purM, purH, pyrC, pyrB, pyrAA, pyrR, pyrP, pyrF, pyrE, pyrG, pdp, tdk*). These results indicate that ComK contributes towards maintaining a steady glucose catabolism response during oxidative stress.

If they are to establish an infection, pathogens such as *S. aureus* must increase their glucose consumption rate[16] under growth

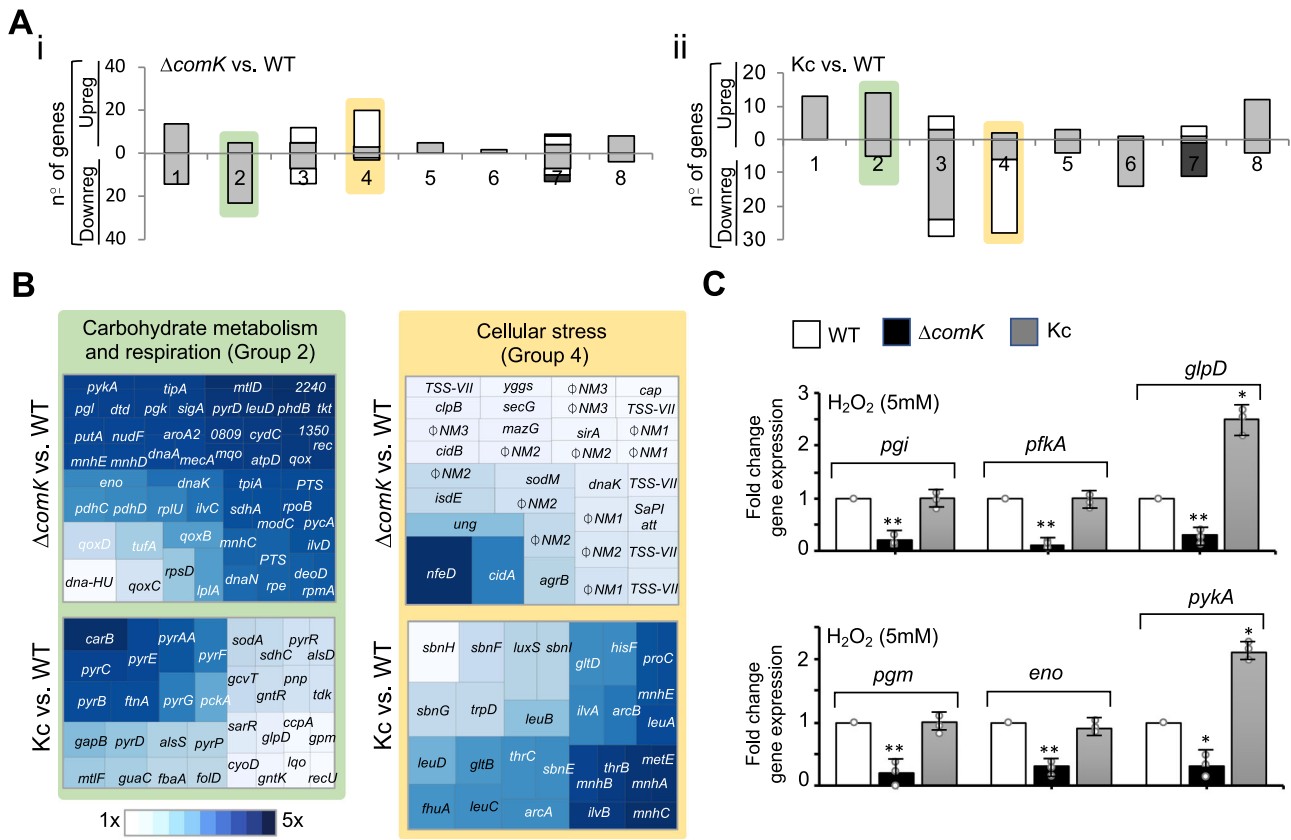

**Fig. 3 ComK induces the expression of glycolytic genes. A** Classification of *comK*-regulated genes using TIGRfam, SEED, and Gene Ontology functional categories. For each category, columns represent the number of regulated genes from DESeq analyses using a fold-change threshold of >2. Pairwise comparisons were made of *S. aureus* strains showing different *comK* expression levels. The Δ*comK* vs. WT comparison (i) involved untreated TSB cultures whereas the Kc vs. WT comparison, and (ii) made use of ROS-induced TSB cultures. Group 1—protein metabolism; group 2—carbohydrate metabolism and respiration; group 3—amino acid (gray) and pigment (white) metabolism; group 4—stress response genes (gray) and phages (white); group 5—virulence and defense; group 6—iron acquisition; group 7—DNA metabolism (gray), cell division (white) and cell envelope (dark green); group 8—other genes. Gene expression changes detected for groups 2 (highlighted in green) and 4 (highlighted in yellow) showed an uniform response under the different growth conditions. Group 2 genes were upregulated in Δ*comK* whereas in the Kc strain they were downregulated. Group 4 genes were induced in the Δ*comK* mutant but downregulated in the Kc strain. **B** Voronoi treemap representing genes regulated by *comK* expression in group 2 (highlighted in green) and group 4 (highlighted in yellow). Genes in which expression was altered are represented and functionally classified. Each section is labeled with the name of the genes represented. **C** qRT-PCR analysis of glycolytic genes in different *S. aureus* strains in $H_2O_2$-treated TSB cultures. *pgi* glucose-6-phosphate isomerase, *pfkA* 6-phosphofructokinase, *glpD* glycerol-3-phosphate dehydrogenase, *pgm* phosphoglucomutase, *eno* enolase, and *pykA* pyruvate kinase. Comparisons were made using one-sided ANOVA with Tukey's test for multiple comparisons; *$p < 0.05$, **$p < 0.01$. Data are shown as mean ± SD of three independent experiments ($n = 3$). Source data are provided as a Source Data file.

conditions in which respiration is inhibited, since fermentative metabolism is fundamentally less energy efficient than respiration. To try to compensate, the glycolytic flux has to be increased[16] (Fig. 4A and Supplementary Fig. 8A). We hypothesized that *comK* induction increases the rate of glycolysis, allowing *S. aureus* cells that are unable to respire to rely on fermentative growth even when glucose availability is low to grow strongly during infection. Consistent with this hypothesis, the expression of key glycolytic genes (Fig. 3C and Supplementary Fig. 8B and C) was found to be higher upon oxidative stress ($H_2O_2$ 5 mM) in the WT strain than in the Δ*comK* mutant. The Kc strain showed a stronger expression of these genes and accordingly, greater concentrations of glycolytic enzymes were detected in cell extracts by mass spectrometry (Supplementary Fig. 8D and E). When glucose abundance was quantified in culture supernatants at different times during the growth cycle, higher concentrations were detected in those from the Δ*comK* strain than from the WT or Kc strains (Fig. 4B), suggesting a reduced glucose consumption rate for the Δ*comK* mutant. Under fermentative growth conditions, glucose can be catabolized to

acetate via the phosphotransacetylase-acetate kinase (*pta-ackA*) pathway. When glucose is in excess, the induction of *cidC* pyruvate oxidase[61] provides an additional pathway to convert pyruvate to acetate[62] (Fig. 4A, Ci). *cidC* expression was repressed in the Δ*comK* mutant growing under oxidative stress conditions ($H_2O_2$ 5 mM), whereas *ackA* expression was not affected (Fig. 4Cii), suggesting that Δ*comK* mutant is less able to import the glucose from the medium. Consequently, the concentrations in the medium of acidic end products ($mM/OD_{600}$), such as acetate and lactate, were lower for the Δ*comK* strain than for the WT or Kc strains (Fig. 4D and Supplementary Fig. 2C), indicating that fermentation is affected in the Δ*comK* mutant in conditions in which respiration is also inhibited by ROS. In agreement, the Δ*comK* mutant showed a greater oxygen consumption rate than either the WT or Kc strains (Fig. 4E), indicating that it relies more on respiration as an energy source (its fermentation rate is lower). This explains the higher toxicity to ROS of the Δ*comK* mutant, thus the lower survival rate of this strain in $H_2O_2$-treated cultures than the WT strain (Fig. 1F). A consequence of using fermentation as energy source in non-respiring bacteria is a fall

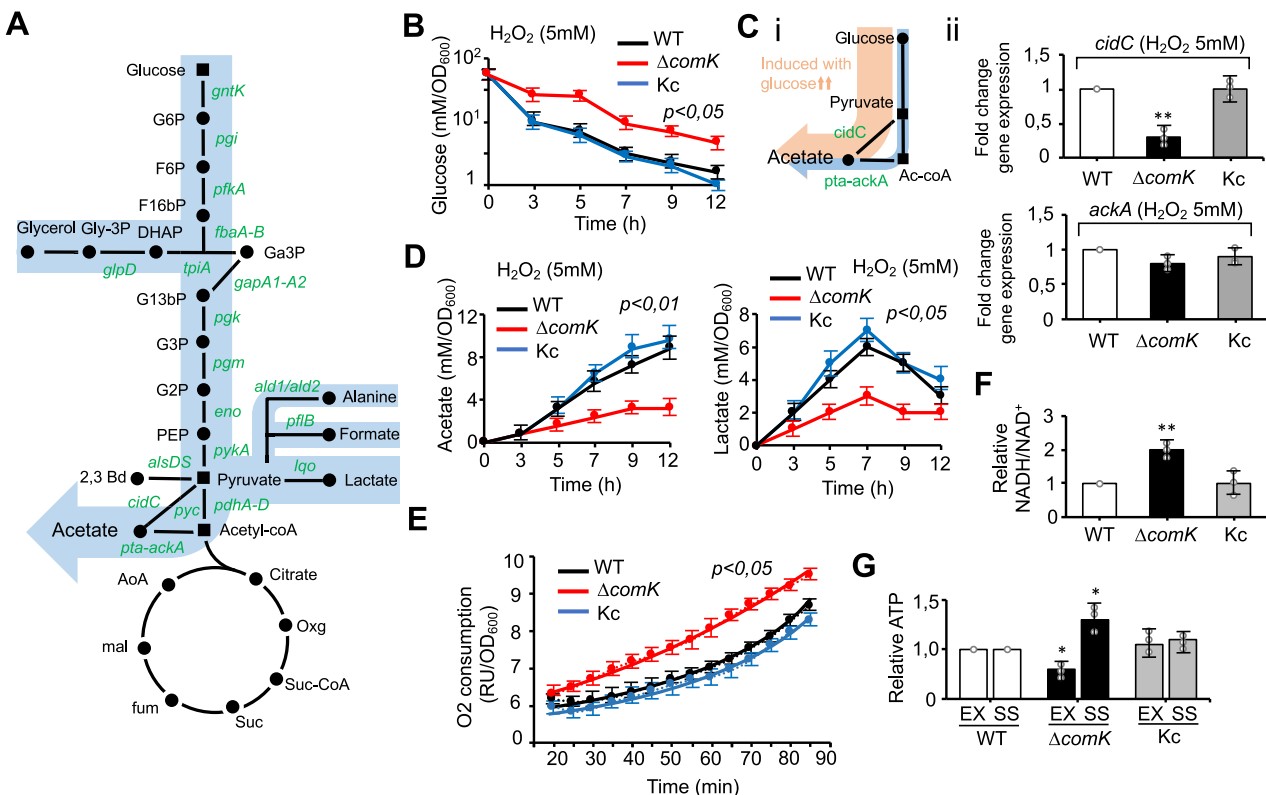

**Fig. 4 ComK upregulates glycolysis in *S. aureus* cells. A** Simplified representation of the *S. aureus* glucose catabolic pathway. Glucose is catabolized to pyruvate, which can be reduced to lactate, 2,3 butanediol (2,3 Bd) or acetyl-CoA. In the presence of high glucose levels, acetyl-CoA is converted to acetate via fermentation (blue arrow) to produce ATP; respiratory chain activity is thus reduced. Lower glucose levels redirect acetyl-CoA towards the TCA cycle. The NADH generated in glycolysis and the TCA cycle is oxidized via the respiratory chain to restore the redox balance and to produce ATP. The genes coding for glycolytic enzymes that are upregulated by *comK* are shown in green. **B** Determination of glucose levels in supernatants of $H_2O_2$-treated cultures. Concentration is represented in relation to culture $OD_{600}$. Statistical significance was measured by one-sided ANOVA with Tukey's test for multiple comparison. Data are shown as mean ± SD of three independent experiments ($n = 3$). **C** Genetic pathway to acetate production. (i) Overflow metabolic pathways to acetate production in *S. aureus*. Under aerobic conditions, acetyl CoA is converted to acetate by the Pta (phosphotransacetylase)/ AckA (acetate kinase) route. Excess glucose activates an additional route that diverts pyruvate towards acetate production via pyruvate oxidase (CidC). (ii) qRT-PCR analysis of *ackA* and *cidC* gene expression in the $H_2O_2$-treated *S. aureus* strains. Differences were examined by one-sided ANOVA with Tukey's test for multiple comparison; **$p < 0.01$. Data are shown as mean ± SD of three independent experiments ($n = 3$). **D** Determination of acetate (left panel) and lactate (right panel) levels in culture supernatants of $H_2O_2$-treated TSB cultures. Concentration is represented in relation to culture $OD_{600}$. Statistical significance was measured by one-sided ANOVA with Tukey's test for multiple comparison. Data are shown as mean ± SD of three independent experiments ($n = 3$). **E** Determination of oxygen consumption rate in $H_2O_2$-treated TSB cultures over time. Statistical significance was measured by one-sided ANOVA with Tukey's test for multiple comparison. Data are shown as mean ± SD of three independent experiments ($n = 3$). **F** Determination of intracellular NADH and $NAD^+$ in the *S. aureus* strains. Results are presented as the $NADH/NAD^+$ ratio for the different strains in relation to the WT. One-sided ANOVA; **$p < 0.01$. Data are shown as mean ± SD of three independent experiments ($n = 3$). **G** Determination of intracellular ATP levels in *S. aureus* cultures collected during the exponential (EX) and stationary phase (SS). ATP levels of the different strains are represented in relation to WT levels. Comparisons were made using one-sided ANOVA with Tukey's test for multiple comparisons; *$p < 0.05$. Data are shown as mean ± SD of three independent experiments ($n = 3$). Source data are provided as a Source Data file.

in reducing potential. In agreement, the $NADH/NAD^+$ ratio remained higher for the Δ*comK* strain than for the WT or Kc strains (Fig. 4F), and the ATP levels were lower in the Δ*comK* mutant than in the WT or Kc strains during exponential growth (Fig. 4G). Overall, these results indicate that *comK* activation enhances *S. aureus* glucose consumption capacity and thus the glycolysis rate in *S. aureus* cells that are unable to respire. This provides *S. aureus* with higher fermentation capabilities to maintain the growth rate in conditions of limited glucose availability.

**ComK-mediated glycolytic induction is associated with enhanced DNA and glucose uptake.** We sought to characterize the ComK-triggered mechanism increasing the rate of glycolysis in non-respiring *S. aureus* to promote fermentation. ComK is a

DNA-binding regulator of gene expression, thus it is possible that ComK binds to and induces the expression of glycolytic genes. It is also plausible that ComK induces the expression of glucose-uptake transport systems. Alternatively, the induction of DNA-uptake may increase the rate of glycolysis in *S. aureus*; in other bacterial species, part of the DNA taken up is commonly degraded into glycolytic intermediates[23,63,64]. To determine whether ComK increases the rate of glycolysis in a direct (by inducing glycolytic genes) or indirect (by inducing DNA- or glucose-uptake genes) manner, we investigated whether ComK binds to key genes involved in these processes. To this end, *S. aureus* ComK was overproduced in a heterologous system and purified to homogeneity (Supplementary Fig. 9A and B). Purified ComK was incubated with PCR-purified 500 bp DNA fragments containing the promoter and upstream sequences of benchmark genes. ComK–DNA interactions were determined using

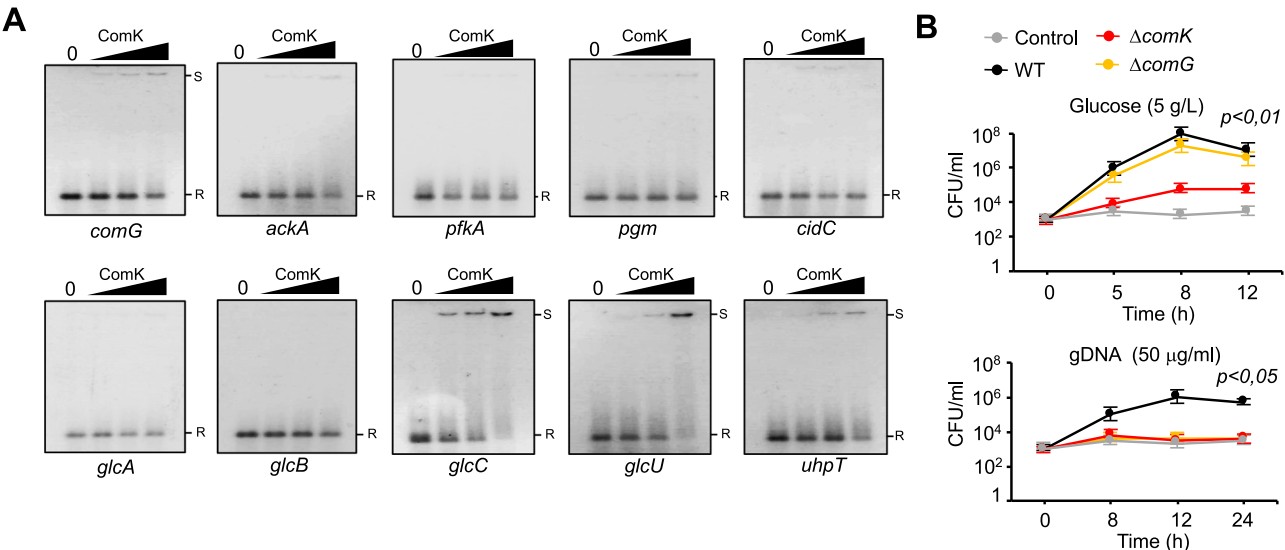

**Fig. 5 ComK binds to and induces the expression of competence and glucose-uptake genes. A** Binding of ComK to regulatory regions of several transcriptional units (*comG*, *ackA*, *glcA*, *glcB*, *glcC*, *glcU*, *uhpT*, *pgm*, *cidC*, *pfkA*). *comG* and *ackA* represent positive and negative controls, respectively. The reference front of unbound DNA is marked with an R. Shifted DNA samples bound to ComK are marked with S. These results are representative of the results obtained in three independent experiments. **B** Growth determination (CFU/ml) in $H_2O_2$-treated SMM cultures (37 °C and 200 rpm agitation) supplemented with glucose or *S. aureus* DNA (gDNA) as a nutrient source. Control sample had no glucose or DNA added. Statistical significance was measured by one-sided ANOVA with Tukey's test for multiple comparison. Data are shown as mean ± SD of three independent experiments ($n = 3$). Source data are provided as a Source Data file.

electrophoresis mobility shift assays (EMSA) (Fig. 5A). *comG* and *ackA* were included as positive and negative reference genes, respectively. *comG* codes for a protein of the DNA-uptake protein machinery that is directly regulated by ComK in *B. subtilis*[65]. *ackA* shows comparable expression levels in the WT and Δ*comK* mutant (Fig. 4Ci). In our assays, the PCR-purified promoter region of *comG* showed a reduced electrophoretic mobility with increasing concentrations of ComK (Fig. 5A), suggesting that ComK binds to *comG* in *S. aureus*. In contrast, no reduced electrophoretic mobility was detected for *ackA*, suggesting that ComK does not bind to *ackA*.

The glycolytic genes *pfkA*, *pgm* and *cidC*, which were upregulated in the Kc strain (Fig. 3B, C), were tested for ComK binding (Fig. 5A). No interaction was detected. Thus, the induction of glycolytic genes may occur indirectly, probably via induction of DNA and/or glucose uptake. To test this, the promoter region of *glcA*, *glcB*, *glcC*, *glcU*, and *uhpT* glucose-uptake transport systems were tested for ComK binding. *glcC*, *glcU*, and *uhpT* showed reduced electrophoretic mobility with increasing concentrations of ComK (Fig. 5A), pointing to the regulation of *glcC*, *glcU*, and *uhpT* expression by ComK. In addition, qRT-PCR validations revealed the downregulation of *glcC*, *glcU*, and *uhpT* genes in the Δ*comK* mutant (Supplementary Fig. 10). These results are consistent with those of previous reports, which argue in favor of the prominent role of *glcC* to the non-respiratory growth of *S. aureus* on glucose[16]. Our results suggest that ComK binds to and induces the expression of genes responsible for DNA (e.g., *comG*) and glucose (e.g., *glcC*, *glcU*, and *uhpT*) uptake systems.

The upregulation of glucose-uptake transport systems by ComK is consistent with the role of ComK in stimulating fermentative growth in *S. aureus* in conditions of poor glucose availability and inhibition of respiration, as it is an infection environment. Nonetheless, ComK upregulates *comG* expression to produce the DNA-uptake protein machinery, which suggests that part of the DNA taken up by *S. aureus* is used as a resource for carbon acquisition or nucleotide acquisition to repair DNA

damage due to oxidative stress[23,63,64]. To investigate this, we generated a Δ*comG* mutant, which is unable to assemble the DNA-uptake protein machinery, and compared the stimulation of growth when respiration is inhibited in WT, Δ*comK* and Δ*comG* cultures using glucose as a nutrient source. Cultures grown in chemically defined minimal medium (SMM)[66] containing glucose (5 g/L) were treated with $H_2O_2$ 5 mM. Final growth rates were calculated by CFU counts. In the presence of glucose, only the Δ*comK* mutant returned a lower CFU count than the WT strain. The growth rate of the Δ*comG* mutant was comparable to that of the WT, suggesting that *comG* is not required for glucose uptake. Using staphylococcal or foreign DNA (from salmon, mouse or *E. coli*) as a nutrient source, the Δ*comG* and Δ*comK* mutants returned much lower CFU counts than did the WT strain although none of the strains were able to produce high growth yields (Fig. 5B and Supplementary Fig. 11A), indicating that *comG* and *comK* play a role in assimilating exogenous DNA for bacterial survival.

We explored the fate of the assimilated DNA using a stable isotope probing (SIP) approach. A DNA fragment of 4.5 kb was generated by PCR using nucleotides containing $^{13}C$ and $^{15}N$ stable isotopes and foreign DNA as a template. Bacterial cultures were supplemented with 20 µg/ml of the labeled DNA and samples were taken at 24 and 48 h incubation period to quantify the incorporation of the $^{13}C$ and $^{15}N$ stable isotopes into the bacterial genome or the proteome. With this approach, it is possible to determine whether the acquisition of exogenous DNA is used as nucleotide source to repair DNA damage (if signal is detected in the bacterial chromosome) and/or as a nutrient source to generate bacterial biomass (if signal is detected in the bacterial proteome) (Fig. 6A). To this end, the genome and the proteome of WT, Kc, Δ*comK*, and Δ*comG* cultures were recovered and the $^{13}C/^{15}N$ content was determined by elemental analyzer isotope ratio mass spectrometry (EA-IRMS) (Fig. 6A). A significant $^{13}C/^{15}N$ signal was detected in the genome of the WT strain compared to a negative control in which the WT strain was grown in the absence of labeled DNA. The genome of the Kc

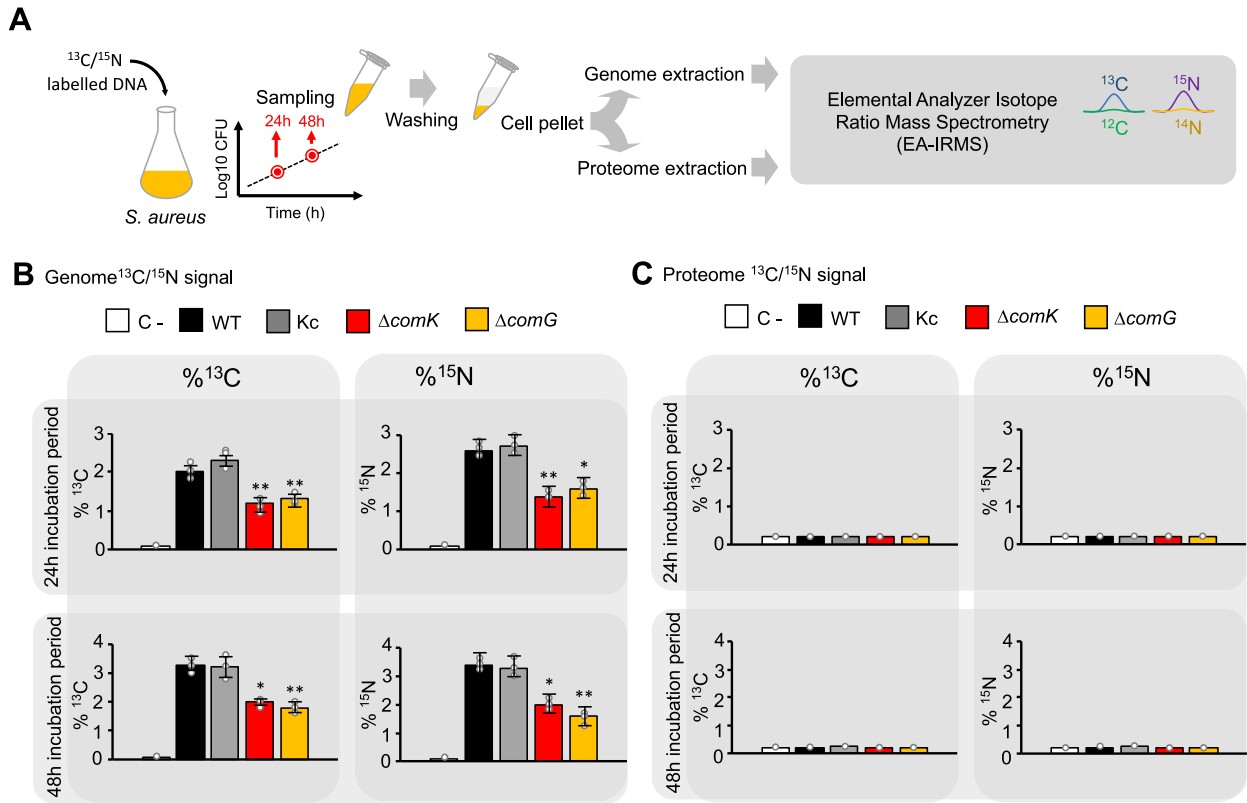

**Fig. 6 ComK induces DNA acquisition as a nucleotide source for DNA repair/synthesis. A** Schematic representation of the stable isotope probing (SIP) experimental workflow using labeled DNA. Staphylococcal cultures were supplemented with a 4.5 kb $^{13}$C and $^{15}$N-labeled DNA fragment, which was PCR-amplified from *B. subtilis* genome (20 μg/ml). Cells were washed, pelleted and the $^{13}$C/$^{15}$N content of the *S. aureus* genome and proteome was analyzed by elemental analyzer isotope ratio mass spectrometry (EA-IRMS). Samples were taken at 24 and 48 h incubation period. **B** EA-IRMS analyses of the $^{13}$C/$^{15}$N content (left/right panel, respectively) of genomes from *S. aureus* WT and different mutants at 24 and 48 h incubation period. **C** EA-IRMS analyses of the $^{13}$C/$^{15}$N content of proteomes from *S. aureus* WT and different mutants. In **B**, **C**, the $^{13}$C and $^{15}$N contents are expressed as atom %. This is the percentage of the total carbon or nitrogen consisting of $^{13}$C or $^{15}$N, respectively. The negative control (C−) was a WT strain grown without labeled-DNA supplementation. As $^{13}$C and $^{15}$N are naturally present in the samples, the $^{13}$C/$^{15}$N basal concentration was used as reference. Comparisons were made using one-sided ANOVA with Tukey's test for multiple comparisons; *$p < 0.05$, **$p < 0.01$. Data are shown as mean ± SD of three independent experiments ($n = 3$). Source data are provided as a Source Data file.

strain also showed higher $^{13}$C/$^{15}$N signal comparable to that of the of WT strain whereas the genome of the Δ*comK* or Δ*comG* mutants showed a significantly reduced $^{13}$C/$^{15}$N signal (Fig. 6B). These results indicate that *comK* induces *comG*-dependent DNA acquisition as nucleotide source to repair DNA damage and probably to synthesize DNA. In contrast, we did not detect a significant $^{13}$C/$^{15}$N signal in the proteome of any strain compared to the negative control (Fig. 6C). Together, these results indicate that DNA acquisition in *S. aureus* is not dedicated to the general formation of biomass, being mainly used as a nucleotide source.

**The Δ*comK* mutant accumulates intracellular ROS and DNA mutations.** During infection, pathogens increase the fermentation rate to counteract the damage to bacterial respiration that is caused by the host ROS-mediated immune response[58,67,68]. Since ComK plays a role in increasing the glycolysis rate when bacterial respiration is inhibited and in the acquisition of DNA to repair the DNA damage caused by ROS, the Δ*comK* mutant is necessarily more sensitive to ROS damage caused by the immune cells (Fig. 1F, G). In agreement with this, transcriptomic analyses and qRT-PCR validations revealed the upregulation of a number of stress-related genes in the Δ*comK* mutant, such as the *spx* and *sarR* stress regulators, those of the Type-VII secretion system, and *sodA* and *sodM* (superoxide dismutases), all of which are usually induced to alleviate oxidative stress[69] (Supplementary Table 3

and Supplementary Data 1). The use of the MitoSOX$^{TM}$ probe showed intracellular O2•− levels to be significantly higher in the Δ*comK* strain than in the WT or Kc strains (Fig. 7A). Consistent with the higher intracellular ROS levels detected in the Δ*comK* mutant, this strain also presented induction of *sodA* and *sodM* (Fig. 7B), underlining that ROS accumulation in this strain causes cell damage.

Pathogens need to protect themselves against the damage that ROS can cause to chromosomal DNA and other cellular components[70]. The upregulation of the *S. aureus* prophage φNM1−4 was detected in the Δ*comK* mutant (Supplementary Fig. 11B). Since prophage induction typically occurs in response to DNA damage[71], we hypothesized that intracellular ROS accumulation in the Δ*comK* mutant would cause damage to the DNA, which is consistent with previous publications showing a link between the accumulation of DNA mutations and the induction of natural competence in close-related pathogens, such as *Streptococcus pneumoniae*[72]. To test this, the accumulation of spontaneous mutations was quantified in WT, Δ*comK* and Kc strains grown in the presence or absence of $H_2O_2$ and subsequently plated on TSB supplemented with rifampicin (Fig. 7C). This identified spontaneous mutations in the *rpoB* gene that confer resistance to rifampicin (Rif$^R$)[73]. In the $H_2O_2$-treated cultures, CFU counts were significantly higher for the Δ*comK* mutant than for the WT and Kc strains, indicating a higher rate of DNA damage in the Δ*comK* mutant. Importantly, a

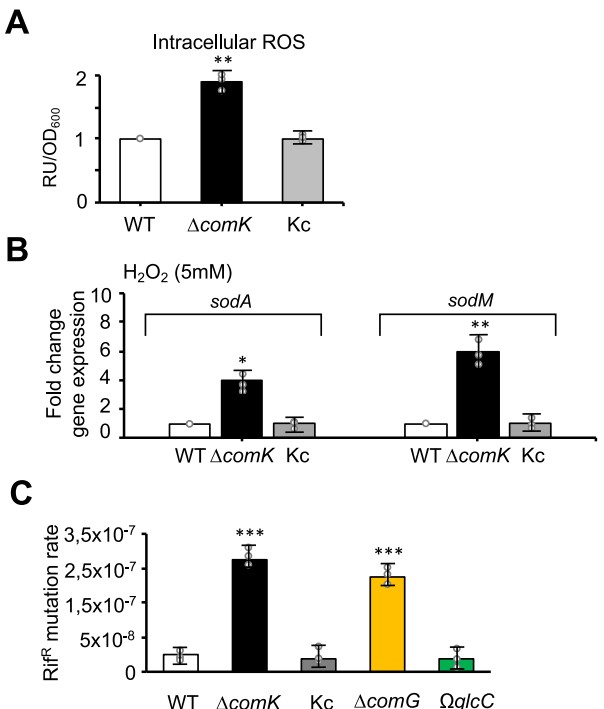

**Fig. 7 The *S. aureus* Δ*comK* mutant showed higher intracellular ROS levels and greater spontaneous DNA mutation rate. A** The Δ*comK* mutant showed higher intracellular ROS levels than WT. Determination of intracellular levels of superoxide radicals using the MitoSOX™ probe in different *S. aureus* mutants. The signal is represented as relative fluorescence units in relation to cultures $OD_{600}$. Comparisons were made using one-sided ANOVA with Tukey's test for multiple comparisons; **$p < 0.01$. Data are shown as mean ± SD of three independent experiments ($n = 3$). **B** qRT-PCR analysis of *sodA* and *sodM* gene expression levels in $H_2O_2$-treated TSB cultures of different *S. aureus* strains. Differences were examined by one-sided ANOVA with Tukey's test for multiple comparisons; *$p < 0.05$, **$p < 0.01$. Data are shown as mean ± SD of three independent experiments ($n = 3$). The Δ*comK* mutant showed higher intracellular ROS levels. **C** Determination of spontaneous mutation rate in different *S. aureus* strains in $H_2O_2$-treated cultures. The mutation rate was determined by quantification of the number of CFU that show spontaneous resistance to rifampicin (50 μg/ml) with respect to the total number of CFU. Differences were examined by one-sided ANOVA with Tukey's test for multiple comparisons; ***$p < 0.001$. Data are shown as mean ± SD of three independent experiments ($n = 3$). Source data are provided as a Source Data file.

comparable rate of spontaneous mutations was detected in Δ*comG* and Δ*comK* mutants, pointing to the role of the competence machinery in DNA damage repair.

***comK* induction is essential for staphylococcal infections to be established**. We previously showed that host cell infection by Δ*comK* mutant is strongly impaired (Fig. 1G). Macrophages respond to *S. aureus* infection by releasing pro-inflammatory signals to coordinate microbial clearance with other immune cells; they also produce anti-inflammatory signals to prevent tissue damage from excessive immune activation[74–76]. We therefore measured the production of pro-inflammatory (TNFα, IL6, and IL1β) and anti-inflammatory cytokines (IL-10) in the supernatant of macrophages exposed to *S. aureus* (Fig. 8A). Macrophages were only exposed to *S. aureus* WT and Kc since the Δ*comK* mutant survives poorly to macrophage phagocytosis. The Kc strain induced greater pro- and anti-inflammatory

responses in macrophages than did the WT, probably due to the constitutive expression of *comK* in this strain, pointing to an association between stronger *comK* expression and a more intense immune response. To investigate the contribution of immune cells, the survival of the different *S. aureus* strains was compared upon incubation with whole blood or plasma (Fig. 8B). A lower CFU count was returned by the Δ*comK* mutant compared to the WT, whereas the Kc strain returned a higher CFU count than the WT. In contrast, all strains showed a comparable bacterial count when incubated with plasma. Thus, the survival of the Δ*comK* mutant is compromised in the presence of innate immune cells.

The importance of *comK* in *S. aureus* virulence was tested using the invertebrate *Galleria mellonella* model. This moth has an innate immune system involving antimicrobial peptides and phagocytic cells[77] and is frequently used to identify bacterial virulence factors that respond to the innate immune system. Three cohorts of 15 larvae received injections of $10^6$ *S. aureus* cells and were incubated at 37 °C for 48 h, after which the number of surviving larvae was counted. Infection of the *Galleria* larvae with the WT strain resulted in a survival rate of ~15% whereas larvae injected with the Δ*comK* mutant ($10^6$ CFU) showed an 80% survival rate ($p < 0.01$), indicating that the virulence of the Δ*comK* mutant is attenuated (Fig. 8C). The survival rate of larvae infected with the Kc strain was comparable to that of the WT strain. Larvae infected with Δ*sigH* or Δ*comG* mutants showed higher survival rate than WT ($p < 0.01$) (Supplementary Fig. 12A). Consistently, the infection of *Galleria* larvae with other *S. aureus* strains, such as USA300 or 8325-4, resulted in a significantly higher survival rate of the larvae injected with the Δ*comK* mutant than those injected with the WT or Kc strain (Supplementary Fig. 12B). In addition, cohorts of BALB/c mice ($n = 10$) were intranasally infected with $10^8$ CFU of the Newman WT, Δ*comK*, or Kc strains (Fig. 8D). Those infected with the WT or Kc strains showed high mortality, with just 20% and 10% surviving 1 and 3 days after infection, respectively. In contrast, mice infected with the Δ*comK* strain showed a survival rate of 90% and 50% after 1 and 3 days, respectively (all $p < 0.01$). Taken together, these results indicate that *comK* expression is crucial for *S. aureus* to establish an infection.

## Discussion

*Staphylococcus aureus* is one of the most successful human pathogens. It causes a variety of serious infections with high mortality and morbidity rates[78]. Its success as a pathogen lies in its ability to metabolically adapt to diverse infection niches, and in the acquisition of resistance to antibiotics by HGT[3]. Natural competence is crucial in promoting HGT, but to date, the involvement of natural competence in bacterial metabolic adaptation is unknown. In this study, we show a link between natural competence and the metabolic adaptation that allows *S. aureus* to successfully infect a host. ComK is produced at very low basal levels under standard laboratory growth conditions but is actively produced in response to infection-related cues, such as ROS and other host damaging agents that inhibit bacterial respiration. In aerobic growth conditions in which carbohydrates are largely available (e.g. exponential growth in TSB), *S. aureus* uses fermentation to grow until the concentration of carbohydrates decreases. Only then do the bacteria rely on respiration to obtain energy[4]. During infection, the ROS and the lower oxygen concentration prevent *S. aureus* from respiring; as such, the bacteria rely on fermentative growth even when glucose availability is low. As glucose availability is limited under infection conditions, an increase in glucose uptake is critical for maintaining effective fermentative growth, which is driven by *comK* induction (Fig. 9A). ComK is a DNA-binding regulatory protein that upregulates the genes coding for the glucose- and DNA-uptake transport

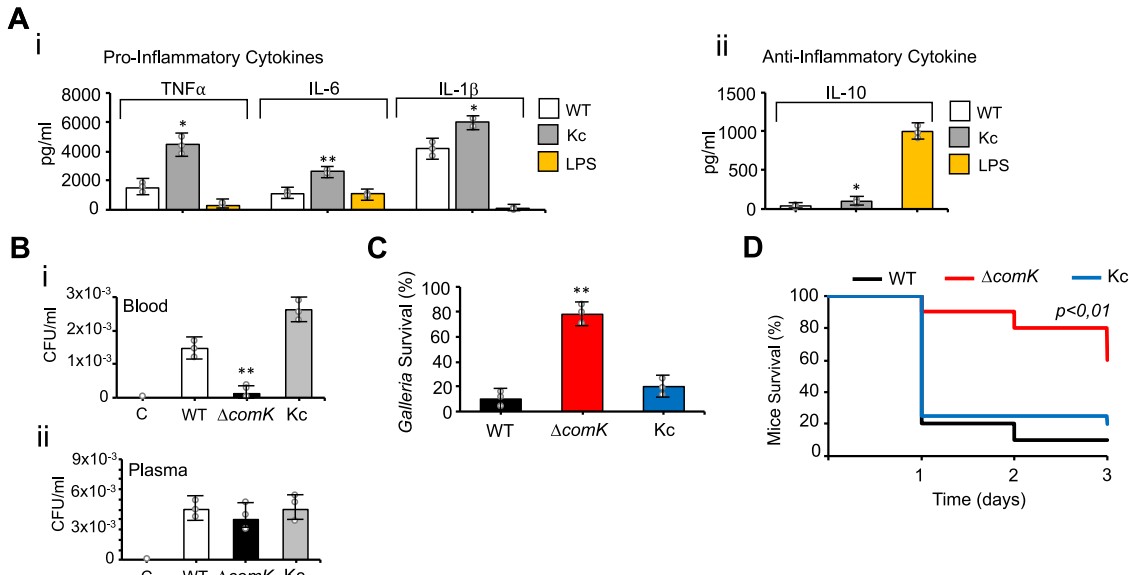

**Fig. 8 *comK* expression is required for *S. aureus* survival during infection. A** Quantification of (i) pro-inflammatory cytokine production (TNFα, IL1β, IL6), and (ii) anti-inflammatory cytokine production (IL10) in the supernatant of monocyte-derived macrophages infected with different *S. aureus* strains (MOI 10) and analyzed at 24 h.p.i. Comparisons were made using one-sided ANOVA with Tukey's test for multiple comparisons; *$p < 0.05$. Data are shown as mean ± SD of three independent experiments ($n = 3$). The activation of the pro- and anti-inflammatory immune response in macrophages is indicative of the severity of infection. Purified lipopolysaccharide (LPS) from *Escherichia coli* was used as a positive control. **B** Bacterial survival, measured as CFU/mL, of the different *S. aureus* strains, after 2 h incubation with non-coagulated human blood (i) or plasma (ii). Differences were examined by one-sided ANOVA with Tukey's test for multiple comparisons; **$p < 0.01$. Data are shown as mean ± SD of three independent experiments ($n = 3$). **C** *Galleria mellonella* were infected with different *S. aureus* strains ($10^6$ CFU). Surviving larvae were counted after 48 h of incubation ($n = 15$ larvae/group; 3 independent experiments). Differences were examined by one-sided ANOVA with Tukey's test for multiple comparisons; **$p < 0.01$. Data are shown as mean ± SD of three independent experiments ($n = 3$). **D** Murine pneumonia model: percentage survival of *S. aureus*-infected mice ($n = 10$). Differences in survival were analyzed by the log-rank test. Statistical significance was measured by two-tailed Student's *t* test, **$p < 0.01$. Source data are provided as a Source Data file.

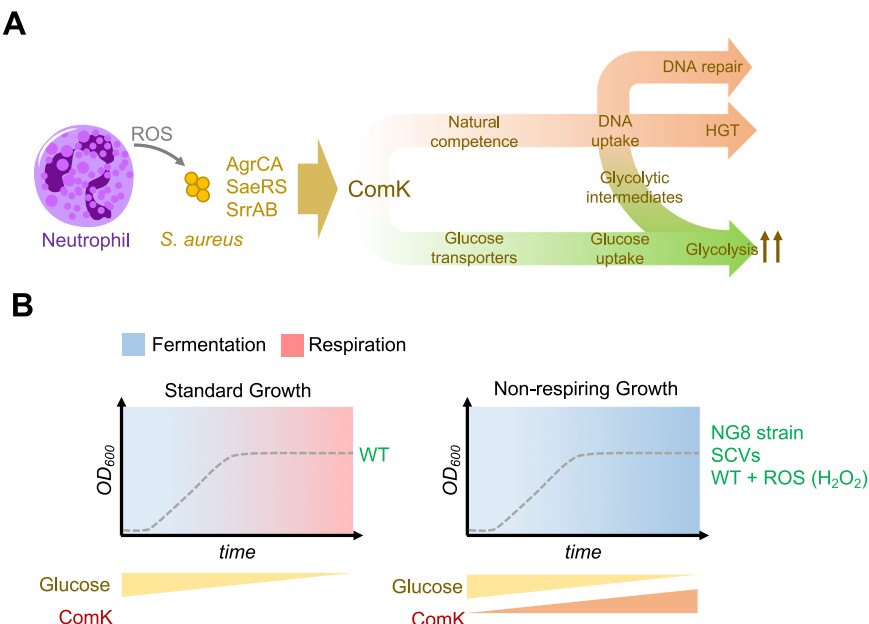

**Fig. 9 A schematic representation of the role of *comK* in *S. aureus*. A** *comK* induction in *S. aureus* laboratory cultures. *comK* expression is not required during the exponential growth of WT cells in TSB medium as carbohydrates are largely available for fermentative growth and when these carbohydrates become more scarce, *S. aureus* relies on respiration to obtain energy. The addition of $H_2O_2$ to the cultures inhibits *S. aureus* respiration; thus, cells grow using fermentation even when the concentration of carbohydrates decreases. In these growth conditions, the cells induce *comK* expression to increase their glycolytic capabilities. **B** *comK* induction during infection. *S. aureus* has little access to oxygen, and bacterial respiration is damaged by the ROS produced by the cells from the immune system (e.g. macrophages and/or neutrophils); respiration rates are therefore reduced. *comK* is induced by the presence of ROS via activation of AgrCA, SaeRS or SrrAB signaling pathways. *comK* induction is critical in cells that are unable to respire, to increase glucose and DNA uptake capabilities and to increase the glycolytic rate that allows and effective fermentative growth. DNA uptake provides an additional source of nucleotides required for the repair of DNA damage caused by ROS and a source of genetic material for horizontal gene transfer.

machineries. This leads to an increase in the glycolytic flux and the glucose consumption rate coupled to an activation of natural competence genes. This allows bacteria that are unable to respire to induce an effective fermentation during the course of infection while increasing their DNA uptake capacity as an additional source of nucleotides to repair the DNA damage caused by oxidative stress and/or to synthesize DNA during cell division. ComK enhances the glycolytic rate by upregulating the expression of genes involved in DNA (e.g., *comG*) and glucose uptake (e.g., *glcC, glcU* and *uhpT*). It may also be that ComK regulates protein levels to induce glycolysis (Supplementary Fig. 8D and E) upon interaction with ClpC protease (Supplementary Fig. 4), as described for *B. subtilis*[45,46].

ComK is induced in response to the respiration inhibiting conditions that are caused by host cell-produced ROS, and low host tissue iron and oxygen levels[6,79] to hinder colonization. However, in laboratory cultures, carbohydrates are quite available and *S. aureus* cells use fermentation to grow until the concentration of carbohydrates decreases; then bacteria rely on respiration to obtain energy and *comK* expression under these conditions is repressed (Fig. 9B). We, however, added $H_2O_2$ to the TSB cultures to inhibit *S. aureus* respiration, similar to what occurs in an infection environment. Under these conditions, *S. aureus* cells use fermentation to grow and when the concentration of carbohydrates decreases, as respiration is inhibited, *comK* expression is induced to increase DNA and glucose uptake, to feed glycolysis to maintain the fermentative growth. The Δ*comK* mutant shows a growth-defective phenotype under these conditions and is strongly impaired in in vitro and in vivo infection models.

The connection between metabolic adaptation and the induction of natural competence may provide important benefits to *S. aureus* during infection. Fermentation induces the secretion of acidic end products (e.g. acetate and lactate) that acidify the medium, making it more toxic to bacterial competitors[64,80]. The DNA of their lysed cells can then be acquired by *S. aureus* cells, perhaps explaining the remarkable genetic variability detected in this pathogen. ComK appears to coordinate the bacterial gene expression program that allows *S. aureus* cells that are unable to respire to take up extracellular DNA to feed fermentative growth and DNA repair/synthesis. In addition, it has been shown that *S. aureus* uses the acidic end products of fermentation as part of a programmed cell lysis mechanism during biofilm formation[57,58]. Extracellular DNA is a component of the biofilm matrix, thus biofilm-associated microbial communities are suitable environments for inducing natural competence and promoting HGT.

While the competence experiments showed DNA uptake to occur in response to *comK* induction when using plasmid DNA, chromosomal recombination events occurred much more rarely, as has been reported in the literature[81]. It may be that recombination events occurred at non-detectable levels, as reported for 'undomesticated' *Bacillus subtilis* strains[82]. Alternatively, it may be that the optimization of nutrient capture is a driving force for the conservation of competence genes in both competent and non-competent firmicutes. In relation to this, we show that part of the DNA taken up during natural competence is used as a nucleotide source to repair the DNA damage caused by oxidative stress or to synthetize DNA during cell division, to promote fermentative growth in respiration-inhibiting growth conditions. This is consistent with the classical behavior of fast-growing cells, such as yeast cells, cancer cells, or bacterial cells undergoing fermentative growth[83] (termed the *Crabtree effect* in yeasts and the *Warburg effec*t in cancer cells) to protect them from ROS damage. In addition to the protective effect of pyruvate against oxidative stress[84], the increased production of pyruvate and other glycolytic intermediates fuels the biosynthetic pathways involved in cell building and generates the nucleotides, amino acids and lipids required by the different cell components involved in cell division[83]. In the context of the present work, it is plausible that *S. aureus* cells induce DNA uptake in conditions of ROS-mediated inhibition of respiration to provide an additional source of glycolytic intermediates as well as additional nucleotides required for cell division or the repair of DNA damage caused by ROS. This is consistent with a number of studies showing that *S. aureus* is a clonal species, in which recombination plays a minor role to genetic variation[85]. In contrast to this, point mutations are significantly more frequent and they are an important driving force to genetic diversification in this pathogen[86,87].

*comK* is conserved in many Gram-positive bacterial species, irrespective of their capacity to induce natural competence (Supplementary Fig. 13). It is thus possible that ComK upregulation of the glycolytic flux is a more general phenomenon in bacteria than previously thought, and may have contributed to the conservation of this gene in competent and non-competent bacterial species. For instance, *Listeria monocytogenes* recovers a functional *comK* gene upon prophage excision during in vitro macrophage infection[29], suggesting that *comK* expression in this pathogen is important during infection, similar to what is reported in this work using the pathogen *S. aureus*. In addition, *comK* expression in *B. subtilis* occurs exclusively in minimal medium rich in glucose as a carbon source[37], suggesting the importance of glucose catabolism in the induction of *comK* activity. Under such conditions, a slight growth defect in the *B. subtilis* Δ*comK* mutant was detected, suggesting that ComK regulates the glycolytic flux in this bacterium as well as in *S. aureus*. Further, it has recently been shown that the subpopulation of *comK*-expressing cells in *B. subtilis* cultures secretes acetate into the growth medium, while the Δ*comK* strain shows reduced acetate production[88]. Thus, the induction of natural competence is connected to the secretion of the acidic end products of fermentation in *B. subtilis* too.

In summary, this work shows that the regulation of bacterial metabolism is essential during infection. Metabolic adaptation enables bacteria to efficiently acquire nutrients while evading oxidative damage caused by the host immune system. The integration of natural competence in the metabolism of bacteria links the mechanisms of DNA repair, genetic variability and bacterial proliferation during infection, offering a more comprehensive view of how metabolic flexibility enables pathogens to colonize the human host.

## Methods

**Bacterial strains, culture conditions, and strain construction**. *S. aureus* strain Newman[89] was used as the WT strain. All in-frame deletion mutants were generated in this genetic background. *S. aureus* RN4220 and *Escherichia coli* DH5α were used only for cloning purposes. A complete list of strains and primers are shown in Supplementary Tables 4 and 5, respectively. If not otherwise specified, *S. aureus* strains were grown in TSB medium or in chemically-defined synthetic minimal medium SMM[66] at 37 °C with agitation at 200 rpm and with no $CO_2$ supplementation. To measure glucose consumption or acetate production, 10 ml of TSB cultures were grown in 100 ml Erlenmeyer flasks and samples were taken at different time points during the growth curve (exponential, post-exponential, stationary, and late-stationary phase). To measure lactate production, 20 ml of cultures were grown in 100 ml Erlenmeyer flasks. When needed, antibiotics were used at the following final concentrations: ampicillin 100 μg/ml, kanamycin 50 μg/ml, streptomycin 100 μg/ml, mupirocin 60 μg/ml, and erythromycin 2 μg/ml. When required, the culture medium was supplemented with a final concentration of 5-bromo-4-chloro-3-indolyl-β-D-galactopyranoside (X-Gal) (40 μg/ml). For DNA uptake experiments, *S. aureus* was grown in SMM medium[66].

Deletion strains were constructed by double-crossover homologous recombination, replacing the gene of interest with a resistant marker cassette (*km*). The Kc strain shows the constitutive expression of *comK* (~10X gene induction) and was constructed by tagging *comK*-ORF with FLAG (by joining PCR), and cloning *FLAG-comK* under the control of an unrepressed xylose promoter into the integrative plasmid pAmy$_{XYL}$[90]. This construct was transferred into the Newman Δ*comK* mutant to create the Kc complemented strain. Constructs were integrated into the *S. aureus* genome by means of double-recombination. After the first recombination step, the constructs were

transferred by Phi11 phage transduction[91] from the RN4220 strain to the Newman strain, in which secondary recombination events were eventually generated.

**Induction of DNA uptake**. DNA uptake by *S. aureus* cells was induced according to a previously published protocol[31]. Overnight cultures of *S. aureus* were grown in TSB medium, and 1:100 dilutions made to inoculate different growth media (TSB, diluted TSB, or SMM medium). Growth occurred at 37 °C with vigorous agitation (200 rpm). Culture samples (500 μL) were collected during the mid-exponential phase and were incubated with 2 μg of genomic DNA or plasmid DNA from *S. aureus*. For plasmid DNA, we used the pMAD[92] derivative plasmid pAMY[90]. The pAMY vector is suitable for plasmid integration into the *amy* chromosomal locus of *S. aureus*. For genomic DNA, we used chromosomal DNA of *S. aureus* with an integration of the pAMY selection cassettes (*erm*+ and *lac*+) into the *amy* locus. Thus, both genomic and plasmid DNA harbor two genetic markers, i.e., an erythromycin-resistance gene (*erm*) and a *lacZ* gene that codes for β-galactosidase (*erm*+, *lac*+) that are derived from the pMAD plasmid[92]. Incubation of the DNA-containing cultures proceeded at 37 °C with mild shaking (60 rpm) for 2 h. Serial dilutions of the cultures were then prepared and CFU counted on agar plates supplemented with erythromycin and X-gal. Positive colonies were confirmed using PCR amplification of *erm* and *lac* genes.

**Cell culture and epithelial invasion assay**. A549 human lung epithelial cells (ATCC CCL-185) were cultured in DMEM supplemented with 10% fetal calf serum (FCS-brand), penicillin (5 μg/ml) and streptomycin (100 μg/ml). Invasion assays were performed by seeding $5 \times 10^4$ cells in 24-well plates and incubating overnight at 37 °C in a 5% $CO_2$ atmosphere. Cells were infected with *S. aureus* Newman at MOI 100, resuspended in 0.5 ml DMEM, and centrifuged at $250 \times g$ for 10 min. Infected cells were incubated for 1 h at 37 °C in a 5% $CO_2$ atmosphere and the culture medium then replaced by DMEM-10% FCS containing 100 μg/ml gentamicin and 5 μg/ml lysostaphin to kill extracellular bacteria. At 2 and 4 h, the cells were collected for RNA isolation.

**Random mutagenesis**. Strains were grown in 10 ml of TSB medium at 37 °C with agitation (200 rpm) until the culture reached stationary phase. Cells were then collected by centrifugation and resuspended in 5 ml of a 0.85% trisodium citrate solution (pH = 5) plus 50 μg/ml of N-methyl N-nitro N-nitrosoguanidine (NG). Cells were incubated for 1 h at 37 °C, washed in PBS buffer, and resuspended in 10 ml of TSB medium. Cultures were incubated at 37 °C with agitation (200 rpm) until they reached twice the initial $OD_{600}$. Serial dilutions were made before plating on TSB agar. Colonies that grew on the agar where monitored for fluorescence using a Bio-rad Chemi Doc system equipped with a fluorescence detector.

**Oxidant susceptibility assays**. Experiments were performed using *S. aureus* cultures grown in TSB medium at 37 °C with agitation (200 rpm). 10 ml of TSB cultures were grown in 100 ml Erlenmeyer flasks and samples were taken at 3, 5, and 7 h after supplementation with $H_2O_2$ 5 mM. $H_2O_2$ 5 mM was added 45 min after inoculation of cultures at $OD_{600} = 0.05$. Serial dilutions were then made and plated on TSB agar for CFU enumeration.

**RNA isolation**. Under the growth conditions explained in greater detail in the main text, cells were harvested and immediately incubated with RNA-protect (Qiagen), or, when derived from cell-line infections, with TRIzol (Thermo Scientific). Total RNA was isolated using the RNeasy kit (Qiagen). To remove all traces of DNA, RNA samples were treated with DNase I (New England Biolabs). RNA purity and quality were determined by Nanodrop (Thermo Scientific).

**Quantitative real-time PCR**. DNA-free RNA samples were used for cDNA synthesis using Superscript III reverse transcriptase (Applied Biosystems) and random hexamer primers. Each PCR was set up in triplicate using the Universal SYBR Green Master mix (Applied Biosystems). The designed primers were used to amplify a 120–200 bp DNA fragment (Supplementary Table 5). Relative amplification was calculated using the $2^{-\Delta\Delta CT}$ Livak Method[93]. The housekeeping *gyrB* rRNA gene was used as an internal reference.

**RNA-seq**. Total RNA was extracted from *S. aureus* Newman (i.e. WT) and Newman Δ*comK* after stress treatment, and TSB cultures of WT and the *comK* complemented strain (Kc) grown to mid-log phase. Total RNA was then extracted using the RNeasy kit (Qiagen). All extracts were submitted to the Otogenetics Corporation (Norcross) for RNA-Seq analysis. The integrity and the purity of total RNA were assessed using an Agilent Bioanalyzer or TapeStation system and by determining the $OD_{260/280}$. rRNA was depleted using the Ribo-Zero Magnetic Gold Kit (Epicentre). cDNA was generated from high-quality total RNA using the SMARTer Universal Low Input RNA Kit (Clontech Laboratories).

The resulting cDNA was fragmented using a Bioruptor (Diagenode), profiled using an Agilent TapeStation system, and subjected to Biomek Fx$^P$ using the Biomek 6000 Automatic Workstation (Beckman Coulter) and the Beckman SPRIworks HT Library Kit (Beckman Coulter) to generate fragment libraries. The manufacturer's instructions were closely followed when performing library

construction. Briefly, following fragmentation the ends were repaired and 'A' bases added to the 3′ end of the fragments. Adapters were then ligated to both ends. The adaptor-ligated templates were further purified using Agencourt AMPure SPRI beads (Beckman Coulter). The adaptor-ligated library was amplified by ligation-mediated PCR and the PCR product purified using Agencourt AMPure SPRI beads once again. Following library construction, a Nanodrop 2000 and an Agilent TapeStation system was used to ensure the quality and quantity of the library. Sequencing was performed using an Illumina HiSeq 2500 system in $2 \times 100$ bp paired-end read mode, according to the manufacturer's instructions. Initial image analysis and base-calling were performed using HiSeq Control Software v.2.0.5 in combination with the real-time analysis (RTA) v.1.17.20.0 program. CASAVA v.1.8.2 was used to generate and report run statistics and the final FASTQ files comprising the sequence information (used in all subsequent bioinformatic analyses). Sequences were de-multiplexed according to the 6 bp index code with 1 mismatch allowed.

**DNA-seq**. Total DNA was extracted from TSB cultures of the *S. aureus* Newman (WT) and Newman NG8 strains. The integrity and purity of the total DNA were assessed using an Agilent Bioanalyzer. Library construction and sequencing was performed using the Illumina HiSeq 2500 system as described above.

**Immunoblotting**. Proteins were separated by SDS–PAGE and transferred to nitrocellulose membranes. In most cases, the protein content was adjusted to 25 μg of total protein per lane using a Nanodrop® Spectrophotometer ND-1000 to quantify the protein concentration of the samples. Immunoblotting was carried out as previously described[90] using a monoclonal antibody against FLAG (1:5000) (Sigma F1804-1MG)) or a polyclonal antibody against 6xHis tag (1:10,000) (Rockland 600-401-382). The secondary antibody Goat anti-Rabbit Ig-HRP (Bio-Rad 172-1019) was added at a 1:20,000 dilution.

**Fluorescence microscopy**. 1 ml of bacterial culture was washed in PBS, resuspended in 0.5 ml of 4% paraformaldehyde solution and incubated at room temperature for 6 min. After two washing steps, samples were resuspended in 0.2 ml of PBS buffer and mounted on a microscope slide. Microscopy images were taken using a Leica DMI6000B microscope equipped with a Leica CRT6000 illumination system. The microscope was equipped with an HCX PL APO oil immersion objective with ×100 1.47 magnification, and a Leica DFC630FX color camera. Linear image processing was performed using Leica Application Suite Advance Fluorescence Software. The YFP fluorescence signal was detected using a 489 nm excitation filter and a 508 nm emission filter (excitation filter BP 470/40, emission filter BP 525/20).

**Bacterial two-hybrid assays**. Bacterial two-hybrid analysis was used to examine the interaction between ComK, MecA, and ClpP. The coding sequences were cloned in-frame into the bacterial two-hybrid expression vectors pKNT25, pKT25, pUT18, and pUT18C (EuroMedex). All combinations of plasmid pairs were co-transformed in *E. coli* BTH101, which harbors a *lacZ* gene under the control of a cAMP inducible promoter. Upon interaction, the T25 and T18 catalytic domains of the adenylate cyclase form an active enzyme leading to the production of cAMP and hence to the expression of the reporter. Protein-interaction assays were performed following the protocol previously described[94]. pKT25-zip and pUT18C-zip, and pKT25 and pUT18C, served as positive and negative controls, respectively. For quantitative measurements of the hybrid signal, β-galactosidase levels were determined. For this, transformants were grown for 48 h at 30 °C in LB medium supplemented with ampicillin (100 μg/ml), kanamycin (50 μg/ml), and 0.5 mM isopropyl-β-ᴅ-thiogalactopyranoside (IPTG; 0.5 mM). The $OD_{600}$ was determined before cells were permeabilized using chloroform and SDS. 200 μl of *o*-nitrophenol-β-galactoside (ONPG; 4 mg/ml) was then added and the reactions stopped after 10 min of incubation at 30 °C by adding 500 μl 1 M $Na_2CO_3$. The $OD_{420}/OD_{550}$ was determined and results recorded in Miller units.

**ComK purification**. A His6 tagged version of ComK was expressed in *Escherichia coli* BL21 Gold (Novagen) strain using the pET28a expression vector (Novagen). ComK expression was induced in *E. coli* cultures (1 l) by using LB media supplemented with 1 mM $MgSO_4$; 0.05% Glucose; 0.2% Glycerol; 0.3% Lactose and 0.5× NPS salts at 18 °C. After 23 h incubation, cultures were harvested and the pellet resuspended in 100 ml of PBS buffer. To induce cell lysis, lysozyme (100 μg/ml) was added followed by cell disruption using a French Press. The cytoplasmic fraction was purified using centrifugation (15,000 × g, 25 min, 4 °C) and the supernatant was loaded into previously equilibrated Ni-NTA resin (Qiagen). Purification was performed using an ÄKTA Pure 25 L chromatography system (GE Healthcare). To remove unbound and non-specific protein, the resin was washed several times with low-imidazole buffer. For elution, 20 ml elution buffer (50 mM Tris–HCl 300 mM KCl pH = 7.5) was added at 250 mM of imidazole; fractions were collected and dialyzed against the final storage buffer (50 mM Tris–HCl 100 mM KCl 50% Glycerol pH 7.5). The concentration of ComK was determined by Bradford assay.

**Electrophoresis mobility shift assay (EMSA)**. To prepare EMSA probes for ComK target genes, the upstream intergenic region up to ~500 bps into the coding sequence of the target gene was PCR-amplified, resolved in agarose gel electrophoresis, and purified. PCR fragments were digested, radiolabeled with [$\alpha$-$^{32}$P] deoxyadenosine 5′-triphosphate using the Klenow fragment enzyme (New England Biolabs) and purified by gel filtration in PD SpinTrap G-25 columns (GE Healthcare). Different concentrations of purified His6-ComK (0, 100, 200, and 400 nM) were incubated with 0.1 nM of radiolabeled DNA in binding buffer (50 mM Tris–HCl 50 mM KCl 0.5 mM DTT 5 mM MgCl$_2$ 0.1 mg/ml BSA pH 7.5) for 15 min at 37 °C. The products where then separated in 0.8% agarose gel using TAE 1X as a running buffer for 3 h at 50 V. The gels were fixed with 7% TCA (v/v) for 15 min, dried and exposed to X-ray film. The retention signal remained in the well upon electrophoresis, likely due to the formation of large DNA–ComK complexes.

**Preparation of SMM medium supplemented with purified DNA**. DNA was isolated using extraction with chloroform:isoamyl alcohol 24:1, as described previously[95]. We obtained DNA from *S. aureus* and *E. coli* cultures, from mouse tail and salmon sperm. Isolated DNA was sonicated (15 cycles of 15 s at 90% cycle and 40% power output). DNA was precipitated with ethanol and resuspended in sterile distilled water immediately before use. *Staphylococcus aureus* was grown in chemically defined synthetic minimal medium SMM[66] overnight. Cultures were pelleted and cells washed and resuspended in 1 ml of PBS buffer. Cell suspensions were used to inoculate 3 ml of SMM cultures at OD$_{600}$ = 0.05. SMM medium contained glucose or DNA as nutrient source at the concentrations specified[23,25]. Cultures incubated overnight at 37 °C with constant agitation (200 rpm). Cell growth was determined by CFU count.

**Stable isotope DNA labeling**. A fragment of $^{13}$C/$^{15}$N-labeled DNA (~4.5 kb) was PCR-amplified from *B. subtilis* genome using $^{13}$C/$^{15}$N-dNTPs (Sigma). The labeled DNA was used to supplement *S. aureus* cultures (20 µg/ml) grown in SSM medium for 24 and 48 h at 37 °C and 200 rpm. Aliquots of 1 ml were collected and washed twice with PBS to obtain genomic DNA or the total protein fraction (using TRIzol from Thermo Scientific). DNA purity and quality were determined by Nanodrop (Thermo Scientific).

**Elemental analyzer isotope ratio mass spectrometry (EA-IRMS)**. DNA or protein samples from the different cultures were placed in tin capsules and were combusted at 1020 °C using an elemental analyzer EA 1112-HT (Thermo Scientific). The elemental analyzer was coupled in continuous flow mode, via the Conflo III interface, to a spectrometer of Isotopic Ratio Masses (IRMS) Delta V-Advantage (Thermo Scientific). As carrier gas, He was used at a flow rate of 95 ml/min. As $^{13}$C and $^{15}$N are naturally present in the samples, the basal concentration of these stable isotopes was quantified in a negative control in which *S. aureus* WT cells were grown in the absence of labeled DNA.

**Blood and plasma killing assays**. *S. aureus* was grown in TSB medium. Cells were pelleted and washed in PBS buffer. $10^6$ cells (50 µl) were resuspended in 150 µl of 5% dilution of human blood or plasma in PBS buffer. Samples were incubated for 25 min at 37 °C. After incubation, the cells were pelleted and washed in PBS buffer. Serial dilutions were plated in TSB medium for CFU enumeration.

**Determination of metabolite concentrations in culture supernatants**. To measure glucose consumption or acetate production, 10 ml of TSB cultures were grown in 100 ml Erlenmeyer flasks containing TSB at 37 °C with agitation (200 rpm). To measure lactate production, 20 ml of cultures were grown in 100 ml Erlenmeyer flasks. Samples were collected at the following time points during the entire growth cycle; exponential (3 h), post-exponential (5 h), stationary (7 h) and post-stationary (15 h). H$_2$O$_2$ was added to bacterial cultures 45 min upon inoculation to a final concentration of 5 mM. Culture supernatants were assessed for glucose, acetate and lactate using determination kits (R-Biopharm). These measure the accumulation of a fluorescence signal, the production of which is dependent on the concentration of the metabolite. Measurements were performed according to previous protocols[96]. To measure oxygen consumption, overnight cultures were harvested and washed in PBS. $10^7$ cells were then resuspended in 150 µl of TSB medium and transferred to the wells of 96-well Costar 3603 plates (black plate with clear bottom). 100 µl of mineral oil were added to each well and the fluorescence signal read according to the manufacturer's instructions. To measure the accumulation of superoxide radicals, 10 ml of overnight cultures were harvested, washed in PBS and diluted at an OD$_{600}$ of 0.1. Bacterial suspensions were washed with PBS and supplemented with MitoSOX$^{TM}$ (ThermoFisher) at a final concentration of 5 µM and incubated for 15 min at 37 °C with shaking. Samples were dispensed into a Costar 3603 96-well plate (black plate with clear bottom) at 200 µl per well. Fluorescence and the OD$_{600}$ were then measured according to the manufacturer's instructions. Intracellular ATP levels were measured using BacTiter-Glo reagent (Promega). *S. aureus* overnight cultures were harvested, washed with PBS and diluted to an OD$_{600}$ of 0.1. 100 µl of bacterial suspension and 100 µl BacTiter-Glo reagent were added to the wells of a Costar 3606 96-well plate. Luminescence was recorded after 15 min of incubation in the dark.

**Rate of mutation to rifampicin resistance (Rif$^R$)**. TSB cultures of the different strains were grown overnight. A sample of $10^9$ cells was plated in TSB agar supplemented with rifampicin 50 µg/ml or TSB agar supplemented with rifampicin and H$_2$O$_2$ 5 mM as an oxidative damage reagent. These plates were then incubated for 24 h at 37 °C. The mutation rate of the different strains was calculated in relation to the H$_2$O$_2$-treated and non-treated samples.

**Phagocytosis and survival in raw macrophages**. Raw macrophage cells were cultured in DMEM supplemented with 10% fetal calf serum (FCS-brand), penicillin (5 µg/ml), and streptomycin (100 µg/ml). Phagocytosis experiments were performed by seeding $3 \times 10^4$ cells in 24-well plates and incubating overnight at 37 °C in a 5% CO$_2$ atmosphere. Cells were infected with *S. aureus* Newman at MOI 25, resuspended in 0.5 ml of DMEM medium, and then centrifuged at $250 \times g$ for 10 min. After 1 h of incubation, the culture medium was replaced by DMEM-10% FCS containing 100 µg/ml gentamicin and 5 µg/ml lysostaphin to kill any extracellular bacteria. Consecutive samples were taken every 24 h for up to 3 days. At every time point, intracellular bacteria were released by cell lysis, adding 500 µl of 0.1% Triton X-100. Following 5 min of treatment, the numbers of bacterial CFU were determined by plating dilutions on TSA plates.

For phagocytosis experiments, bacteria were added to monocyte-derived macrophages (MDM) cultures for 30 min at a ratio of 10:1 bacteria-to-MDM. These cultures were then washed twice with DMEM supplemented with 1% penicillin–streptomycin (Invitrogen) and 1 mg/ml of lysostaphin (Thermo Scientific) to eliminate non-internalized bacteria. At different time points, the MDM were lysed using Triton 0.5% for 10 min. Lysates were seeded on agar plates and CFUs enumerated.

**Human monocyte-derived macrophage isolation and cell culture**. MDM were obtained from peripheral blood mononuclear cells (PBMCs) isolated from the buffy coat fraction by Ficoll-Plus gradient centrifugation (GE Healthcare Bio-Sciences) and monocytes seeded. Monocytes were cultured in DMEM medium supplemented with 10% FBS (Invitrogen) in a humidified 5% CO$_2$ atmosphere at 37 °C incubator for 2 weeks. Culture media were renewed every 3 days and the cells monitored morphologically for differentiation. All of the reagents used for cell culture were endotoxin-free, as determined using the Limulus amebocyte lysate test (Cambrex).

**Inflammatory cytokines quantification**. The cytokine levels in the MDM culture supernatants were determined using the Cytometric Bead Array Flex Set (BD Biosciences) following the manufacturer's protocol. Data were collected by flow cytometry using a BD FACSCalibur flow cytometer and analyzed using FCAP Array Software v3.0 (BD Biosciences).

**Animal infection models**. For infections of wax moth (*Galleria mellonella*), larvae were injected with $10^6$ *S. aureus* CFUs in a MgSO$_4$ 10 mM solution and incubated at 37 °C for 48 h. Three independent experiments were performed with 15 larvae per tested strain and experiment. The control group was injected with 10 mM MgSO$_4$. Comparisons of survival were made using the two-tailed Student *t* test. Significance was set at $p < 0.05$.

Inbred 16 weeks old female mice BALB/c (Charles River Breeding Laboratories) weighing 20–24 g were also used for infection experiments (pneumonia model). Animals ($n = 10$ per group) were infected intranasally with $10^8$ CFU suspended in 20 µl of saline solution. Survival was recorded over 3 days. Severe weight loss (>20%) was deemed the experimental end point for each animal. Differences in survival between the different strains were examined using the log-rank method, employing GraphPad Prism v.7 software.

**Ethics statement**. All experimental animal studies were approved by the Committee on Ethics in Animal Experiments of the Government of Lower Franconia (55.2-2532-2-57) and were performed in strict accordance with the guidelines for animal care and experimentation set forth in the German Animal Protection Law, and in EU Directive 2010/63/EU. Animal maintenance/housing was conducted in individually ventilated cage (IVC) systems in order to maintain disease-free conditions. The following standard conditions are used: 12 h light/dark cycle, 21 ± 1 °C and 55 ± 5% relative humidity. Rodent chow and water are available ad libitum. Animal changing-working stations were available with pre-filtered air and negative pressure to reduce the risk of infection. Every 3 months, hygienic and health monitoring was performed in the animal facility, in accordance to FELASA (Federation of European Laboratory Animal Science Associations) recommendations. Prior animal experiments, rodents are generally allowed to rest for, at least a week, to adapt them to the experimental animal facility. All animals were sacrificed at the end of the experiment by CO$_2$ inhalation.

**Statistical methods**. Data are shown as mean ± SEM of three independent biological replicates, each including three technical replicates. Data presented for immunoblotting, fluorescence microscopy assays or EMSA are representative results of three biological replicates. Statistical analyses were performed using GraphPad Prism software v.7, employing the statistical methods indicated in figure

legends. Significance was set at $p < 0.05$. Pairwise comparisons were performed using the two-tailed Student $t$ test. Analysis of variance (one-sided ANOVA) was performed to compare differences among groups of means.

**Reporting summary**. Further information on research design is available in the Nature Research Reporting Summary linked to this article.

## Data availability

The demultiplexed and coverage files that support the RNAseq and DNAseq findings were deposited in the NCBI's GEO database (GSE155016) and in the NCBI's SRA database (SRR12329121), respectively. Source data including uncropped gel images are provided with this paper. Source data are provided with this paper.

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

## Acknowledgements

The authors thank Ana Marina Cabrerizo (CNB-CSIC, Spain) and Isa Westedt (University of Würzburg, Germany) for technical assistance, and Adrian Burton for language and editing assistance. We thank Vanessa Peiró (Autonomous university of Madrid) for assistance with EA-IRMS. This work was funded by grant PID2020-115699GB-I00 (MINECO, Spain) and ERC 335568 (European Research Council, EU) to D.L., and by the ERA-NET Infect-ERA StaphIN PCIN-2015-151 (MINECO, Spain) and 031L0094 (BMBF, Germany) to D.L. and A.E., respectively.

## Author contributions

M.C., J.G.-F., I.C.A., A.Y., J.A.-O., C.L., B.O., and D.L. performed most of the experiments. M.C., J.G.-F., I.C.A., A.Y., J.A.-O., C.L., B.O., K.O., E.L.-C., K.U.F., A.E., and D.L. undertook the formal analysis of the results. D.L. wrote the original draft of the paper and J.G.-F., E.L.-C., K.O., K.U.F., and A.E. assisted with review and editing.

## Competing interests

The authors declare no competing interests.
