## [Peer Review File · Nature Communications]

Reviewers' Comments:

Reviewer #1:

Remarks to the Author:

The manuscript entitled "The Induction of Natural Competence Adapts Staphylococcal Metabolism to Infection" by Cordero, M. et al. describes a role of the ComK transcription factor in modulating glycolytic flux as well as nucleic acid uptake during times of respiratory distress. The authors suggest that in response to host inflammatory ROS, bacterial respiration is inhibited thereby inducing the expression of ComK, which targets glucose transporters thereby maximizing glycolytic flux. This is said to be necessary given the inherent decrease in energy derived from fermenting carbohydrates as opposed to cellular respiration. The data are well controlled for the most part and clearly presented. Only a few minor issues stand out upon review:

1. The major issue is the use of the terms fermentation versus respiration. One could think of the difference between the two depends on the use of the F₁F₀ ATPase. If this complex uses PMF to generate ATP, then the cells are respiring. If it hydrolyzes ATP from substrate level phosphorylation to generate PMF, the cells are fermenting. This was never really assessed so the terms seem to be used erroneously at times. For instance, in Figure 7, at low cell densities like those at the beginning of a growth curve, the cells are likely respiring. Later on at high cell densities, oxygen may be used faster than it can dissolve into the medium, therefore many of the cells are forced to ferment.
2. Given the above, however, it seems odd that cells are synthesizing lactate so early in the growth curve (Figure 2Fii). Normally, *S. aureus* converts glucose to acetate early on and only later does lactate production ensue. This raises concerns with the WT Newman (a strain known to have a constitutive Sae system).
3. Line 261, the authors quote biochemistry text books for the 38 ATPs made from respiring on glucose. However, *S. aureus* converts glucose to acetate yielding 4 ATP as opposed to 2 when fermenting. Many strains cannot reacquire and oxidize the acetate later on.
4. The introduction seems to be a bit biased towards HA-MRSA infections, given that after 2000, many more infections are of the CA-MRSA etiology.
5. Line 184, the word "and" should not be italicized.
6. Since a Δ comG mutant is represented in figures, merely signifying the Δ comK mutant as " Δ " can be confusing. Δ comK should be used instead.
7. Figure 4, panel A is duplicated in the supplemental information.
8. Figure 2 panel B indicates pcomG::GFP, but in the text YFP is mentioned.
9. Figure 5A, none of the EMSAs contain a competitive DNA fragment to rule out non-specificity. This is only seen as a minor issue as so many of the EMSAs were negative.

Reviewer #2:

Remarks to the Author:

In the paper «The induction of natural competence adapts staphylococcal metabolism to infection», the authors used a variety of methods to improve the understanding of the function of comK in *S. aureus*. This is of relevance, given the role of *S. aureus* in hospital-associated infections and the increasing antibiotic resistance levels.

Cordeiro et al identified *S. aureus* growth conditions that led to the expression of the ComK regulator. These were conditions that *S. aureus* are likely to encounter during host infection, including growth on human alveolar epithelial cells, exposure to macrophages, H₂O₂, and use of *Galleria* and mouse infection models. Further, the authors investigated possible mechanisms behind comK activation/repression. They found that although MecA bound to ClpP, the complex did not promote ComK degradation. Deletion of the genes for two sigma factors, SigH and SigB, and four other regulators resulted in mutants with reduced comK expression in response to H₂O₂. In a mutagenized strain, increased expression of comK was associated with increased fermentation. Small colony variants exhibited higher levels of comK, thus supporting the association between increased comK expression and defective respiration.

The authors found that the expression of key glycolytic genes by the WT in the presence of H₂O₂ were higher than in the comK mutant. Mass spectrometry and other biochemical assay results supported the findings showing the association of comK expression with reduced respiration and

increased fermentation. ComK did not bind to the promoter of glycolytic genes that were differentially expressed. On the other hand, it bound to the comG promoter and glucose uptake systems, suggesting a direct regulatory role in uptake of DNA and glucose. The authors also found a higher rate of DNA damage in the comK mutant in response to growth in the presence of H₂O₂. ComK expression was associated with increased infection of macrophages, survival in blood, and virulence in *G. mellonella* and mouse models of infection.

My general impression is that the authors have done a lot of work in getting an overall picture of the role of comK, but it did not go deep enough in some of the addressed questions to support some of the main conclusions. The debate on whether nutrition is the purpose of competence has been going on for decades, illustrating the difficulties in finding a straight answer. Most of the evidence indicates, however, that transformation is the main purpose. There are good reviews on this, including Johnston, C., Martin, B., Fichant, G. et al. Bacterial transformation: distribution, shared mechanisms and divergent control. *Nat Rev Microbiol* 12, 181–196 (2014). The conservation of genes for protection of single strand DNA in *S. aureus* indicates that this is also the case in this species (ref 35).

The authors conclude on l. 114 to l. 116 that their work “demonstrates that natural competence is conserved in diverse bacterial species beyond fostering genetic variability, specifically to provide bacteria with additional nutritional and metabolic possibilities”. Since the study focused on *S. aureus*, the authors should be careful in extending the conclusion to other bacterial species. While the effector genes are conserved in different species (DNA binding, uptake, etc), there is a general consensus that the environmental cues triggering competence show high variability among competent species. I also think that some points would need to be addressed before arriving to the conclusion on whether this is the case for *S. aureus*. These are specified below, together with other points.

(1) While the association between oxidative stress and competence described by the authors is of relevance since this has not been described for *S. aureus* before, this is not unique to *S. aureus*. In *S. mutans*, for instance, oxidative stress has different effects on the competence signaling pathway (De Furio M, Ahn SJ, Burne RA, Hagen SJ. Oxidative Stressors Modify the Response of *Streptococcus mutans* to Its Competence Signal Peptides. *Appl Environ Microbiol.* 2017 Oct 31;83(22):e01345-17). Not surprisingly, the mechanisms seem to differ, and this is a point that could be addressed in the paper.

(2) Growth curves for the different conditions and mutants need to be presented, since some of the differences can be due to collection of data when the bacterial cultures were in different phases, or due to a slow-down in growth. For instance, results on acetate and lactate production can be affected by growth rate, since these represent cumulative values and not real-time production rates. The same is valid for transcriptome data, which is highly influenced by growth phases. Small differences in growth rates can have a large impact on the transcriptome.

(3) Several of the figures show fold change in comK expression. However, comparison with a gene that is not expressed at all, as reported in l. 127 “we were unable to detect comK expression under laboratory growth conditions”, should give a fold change of 0.

(4) Would the authors confirm that the number of samples and replicates was the same for all different experiments, including infection models, macrophage assay, blood/plasma growth, transcriptome, qPCR, protein interactions, etc, as described in the methods? I suggest this information should be presented for each of the figures.

(5) A statistically significant effect in ANOVA is generally followed by additional tests to assess to specific differences. I believe the authors have done it, but this information is not presented in the text.

(6) The conclusion by the authors that DNA uptake is used for nutrition is not based on direct evidence, but differences in growth in the presence of DNA substrate. As an example, differences in chain formation may affect CFU counts, and pleiotropic effects due to mutations can not be excluded. The fate of DNA needs at least to be investigated in DNA binding and uptake assays, usually with labelled DNA. L.344

Other points:

l. 323 Binding to the comG promoter has been reported before (ref 35)

l. 379 comK mutant does not survive macrophage assay- but how to know without measuring it?

l. 380 Should not WT and Kc respond in the same way, since ROS should in theory activate the system in the WT?

l. 385 Blood and plasma should vary in other factors than only blood cells.

-introduce SigH-synergy with comK literature

- l. 187- rswB deletion mutant had an effect that is presented only in the figure.
- L 102 and -l.123- Ref 35 is not correct –DNA transfer and DNA binding were not observed in comK expressing strain
- The use of anti-oxidants would have strengthened the conclusion that ROS activate the competence response (eg catalase, parakat)
- How do the authors explain the finding that the degree of aeration had no effect on comK expression? L. 134.
- l.l. 96 Reference for *L. lactis* competence is lacking
- DNA mutations are also known to trigger competence in *S. pneumoniae* (Gagne AL, Stevens KE, Cassone M, et al. Competence in *Streptococcus pneumoniae* is a response to an increasing mutational burden. PLoS One. 2013;8(8):e72613). Authors could mention it.
- Could production of ROS by *S. aureus* itself contribute to the observed effects?
- How does the transcriptome results compare to those presented in ref 35?
- l.213- "a large number": please specify it
- l 198- introduce comG function
- Fig 6. % survival results: could it be inhibition of growth, rather than death?
- Fig 6 (B) Was there growth or death in blood and plasma? Usually in these assays a mutant that can not divide in blood is used, to exclude growth as a confounding factor.
- NG8 strain: it would help the reader if the strain background is explained in the figure legend
- Could the NG8 phenotype showing changes in acetate, lactate and O₂ consumption be a result of an altered growth rate? I miss a growth curve for NG8, and also the other strains, particularly in the presence of H₂O₂.
- l.254- a Venn diagram showing the common genes increased in the Kc vs WT and reduced in the comK deletion vs WT would be helpful.
- It would be a valuable information to know whether comK is expressed in vivo. Have the authors measured it?
- Fig. 7. The information in the legend does not match with the figures. Looks like A and B is inverted.
- Fig. 3. L 1066-1067: looks like the information in the legend is inverted.
- fig 3 A and B, the comparison description in the figures is different from the description in the legend (WT vs delta, delta vs WT)

Reviewer #3:

Remarks to the Author:

The function of genes in many gram positive species, including *S. aureus*, that are orthologous to those related to the induction of competence and transformation in *Bacillus subtilis* and *S. pneumoniae* are not well established. The investigative team has found that the competence pathway in *S. aureus* is induced (via ComK) under oxidative stress and that a comK allelic replacement mutant is less virulent in both murine and *Galleria* infection models. This is an original and impactful observation. Previous literature has documented the ability of *S. aureus* to facilitate DNA transformation upon induction of SigH (which was comG dependent).

Typically comK is not induced during growth in rich medium such as TSB. However, upon induction of comK, *S. aureus* grows via a fermentative mechanism. Appropriate experiments were performed to document this (acetate, lactate, NAD/NADH ratio, oxygen consumption). Further, it was determined that induction of comK increases glucose consumption, it is probable that this is via direct binding of ComK and subsequent induction of genes encoding glucose transporters. Thus, the evolving model proposed by the investigators is that upon oxidative stress, comK is induced and one outcome is that glucose transporters are induced further increasing glycolysis as respiration is inhibited (presumably via the oxidative stress). As such, comK is required for infection in both murine and *Galleria* models.

However, although this is novel and impactful, the investigators did not necessarily define how the competence network and DNA uptake is important for the phenotypes described above. Clearly, this is the most impactful question as glucose catabolism is known to be very important for initial *S. aureus* infection. It seems that the investigators need to determine if upon comK induction whether ComG-dependent DNA acquisition are required for nucleotide acquisition, carbon

acquisition, or required to fix DNA damage due to oxidative stress. Experimental approaches to address these questions are not necessarily difficult to address ($^{13}\text{C}/^{15}\text{N}$ experiments or transformation experiments in defined media lacking nucleotides). Further, to delineate if nucleotide acquisition is required for the phenotypes associated with the animal models of infections one could utilize the comG mutant with or without comK mutation. Since ComK function to induce glucose acquisition, the lack of virulence would be predicted.

Below are some specific comments.

1. Previous manuscripts have documented that the competence system is induced via SigH. This aspect was not necessarily addressed in this manuscript. Is SigH required for any of the phenotypes discussed in the manuscript? This is important to note one way or the other.

2. Line 459 and 460. The authors state that they documented that DNA is used for a nutrient source? Not sure this has been documented nor have the authors documented whether DNA can act as a carbon source (or even a nitrogen source) in *S. aureus*.

3. In the materials and methods section, the transformation experiments were not well described. What was the plasmid used? Where was lacZ inserted in the chromosome and what lacZ gene was utilized. Further, Morikawa and colleagues utilized 10 micrograms of chromosomal DNA (typically has to be spooled to generate this concentration). This may be why the authors could not detect transformants using chromosomal DNA.

4. Lastly, if chromosomal DNA is acquired and used as a source to repair DNA, one would expect to detect this process using population biology techniques. These studies to date have found that *S. aureus* does not evolve through recombination. This should be discussed in the discussion section.

Reviewer #1 (Remarks to the Author):

The manuscript entitled “The Induction of Natural Competence Adapts Staphylococcal Metabolism to Infection” by Cordero, M. et al. describes a role of the ComK transcription factor in modulating glycolytic flux as well as nucleic acid uptake during times of respiratory distress. The authors suggest that in response to host inflammatory ROS, bacterial respiration is inhibited thereby inducing the expression of ComK, which targets glucose transporters thereby maximizing glycolytic flux. This is said to be necessary given the inherent decrease in energy derived from fermenting carbohydrates as opposed to cellular respiration. The data are well controlled for the most part and clearly presented. Only a few minor issues stand out upon review:

1. The major issue is the use of the terms fermentation versus respiration. One could think of the difference between the two depends on the use of the F1FO ATPase. If this complex uses PMF to generate ATP, then the cells are respiring. If it hydrolyzes ATP from substrate level phosphorylation to generate PMF, the cells are fermenting. This was never really assessed so the terms seem to be used erroneously at times. For instance, in Figure 7, at low cell densities like those at the beginning of a growth curve, the cells are likely respiring. Later on at high cell densities, oxygen may be used faster than it can dissolve into the medium, therefore many of the cells are forced to ferment.

We have performed a revision of the text according to this referee's suggestion and clarified the terms fermentation/respiration in this revised manuscript (lines 81-88 of the Introduction section and lines 464-467 of the Discussion section). In aerobic growth conditions in TSB medium, in which carbohydrates are quite available (e.g. exponential growth in TSB cultures), *S. aureus* uses fermentation to grow until the concentration of carbohydrates decreases. Only then do the bacteria rely on respiration to obtain energy. Such favorable growth conditions do not require additional glucose uptake mechanism to feed fermentation, thus *comK* is repressed. During infection, ROS and low oxygen prevent *S. aureus* from respiring; the bacteria rely on fermentative growth even when glucose availability is low. An increase in glucose uptake is critical for maintaining effective fermentative growth, especially when glucose availability is low, which is driven by *comK* induction. In our assays, we added H₂O₂ to the TSB cultures to inhibit *S. aureus* respiration. Cells use fermentation to grow and when the concentration of carbohydrates decreases, *comK* expression is induced to increase glucose uptake and thus maintain fermentative growth, similar to that occurring during infections. Accordingly, the $\Delta comK$ mutant shows a growth-defective phenotype under these conditions. We quantified NADH/NAD and ATP concentration, oxygen consumption or the accumulation of fermentation-end products, to determine the contribution of fermentation or respiration to bacterial growth.

2. Given the above, however, it seems odd that cells are synthesizing lactate so early in the growth curve (Figure 2Fii). Normally, S. aureus converts glucose to acetate early on and only later does lactate production ensue. This raises concerns with the WT Newman (a strain known to have a constitutive Sae system).

In the revised manuscript, we have expanded our description of the growth conditions and provided additional information regarding the lactate/acetate measurements (lines 559-562 and 768-770). To perform these measurements, we followed the growth conditions for acetate/lactate measurements described in Somerville and Proctor, 2013 (PMID: 23324109) and Vuong et al., 2005 (PMID: 15838022). As reported, we used different growth conditions to optimize acetate or lactate production in the cultures thus, acetate and lactate production levels are not interrelated in this assay. In this regard, the strain Newman's behavior is comparable to that of other staphylococcal strains described in the literature.

3. Line 261, the authors quote biochemistry text books for the 38 ATPs made from respiring on glucose. However, S. aureus converts glucose to acetate yielding 4 ATP as opposed to 2 when fermenting. Many strains cannot reacquire and oxidize the acetate later on.

We have appropriately modified this sentence (line 284).

4. *The introduction seems to be a bit biased towards HA-MRSA infections, given that after 2000, many more infections are of the CA-MRSA etiology.*

We have included additional information about HA-MRSA and CA-MRSA infections in the introduction section according to the referee's recommendation (lines 55-56).

5. *Line 184, the word "and" should not be italicized.*

We have replaced this sentence with a more detailed description of the role of SigH in staphylococcal competence (lines 190-203).

6. *Since a $\Delta comG$ mutant is represented in figures, merely signifying the $\Delta comK$ mutant as " Δ " can be confusing. $\Delta comK$ should be used instead.*

We have replaced the labelling Δ with $\Delta comK$ in this revised manuscript, as suggested by this referee.

7. *Figure 4, panel A is duplicated in the supplemental information.*

An updated version of Supplemental Figure 5 is included in this revised manuscript.

8. *Figure 2 panel B indicates $pcomG::GFP$, but in the text YFP is mentioned.*

We have corrected this figure accordingly.

9. *Figure 5A, none of the EMSAs contain a competitive DNA fragment to rule out non-specificity. This is only seen as a minor issue as so many of the EMSAs were negative.*

We did not include a competitive DNA fragment as control of the EMSA assay because the EMSA included a positive (*comG*) and a negative (*ackA*) control assays to confirm the specific binding of ComK. As this referee stated, a competitive DNA fragment is typically used when all samples show a positive binding signal, to rule out that the signal is due to unspecific binding. In our assay, many samples resulted negative, which further indicates that ComK binds to only certain promoters, likely in a specific manner. We have clarified this in this revised manuscript (lines 329-335).

Reviewer #2 (Remarks to the Author):

In the paper «The induction of natural competence adapts staphylococcal metabolism to infection», the authors used a variety of methods to improve the understanding of the function of comK in S. aureus. This is of relevance, given the role of S. aureus in hospital-associated infections and the increasing antibiotic resistance levels.

Cordeiro et al identified S. aureus growth conditions that led to the expression of the ComK regulator. These were conditions that S. aureus are likely to encounter during host infection, including growth on human alveolar epithelial cells, exposure to macrophages, H₂O₂, and use of Galleria and mouse infection models. Further, the authors investigated possible mechanisms behind comK activation/repression. They found that although MecA bound to ClpP, the complex did not promote ComK degradation. Deletion of the genes for two sigma factors, SigH and SigB, and four other regulators resulted in mutants with reduced comK expression in response to H₂O₂. In a mutagenized strain, increased expression of comK was associated with increased

fermentation. Small colony variants exhibited higher levels of comK, thus supporting the association between increased comK expression and defective respiration.

*The authors found that the expression of key glycolytic genes by the WT in the presence of H₂O₂ were higher than in the comK mutant. Mass spectrometry and other biochemical assay results supported the findings showing the association of comK expression with reduced respiration and increased fermentation. ComK did not bind to the promoter of glycolytic genes that were differentially expressed. On the other hand, it bound to the comG promoter and glucose uptake systems, suggesting a direct regulatory role in uptake of DNA and glucose. The authors also found a higher rate of DNA damage in the comK mutant in response to growth in the presence of H₂O₂. ComK expression was associated with increased infection of macrophages, survival in blood, and virulence in *G mellonella* and mouse models of infection.*

*My general impression is that the authors have done a lot of work in getting an overall picture of the role of comK, but it did not go deep enough in some of the addressed questions to support some of the main conclusions. The debate on whether nutrition is the purpose of competence has been going on for decades, illustrating the difficulties in finding a straight answer. Most of the evidence indicates, however, that transformation is the main purpose. There are good reviews on this, including Johnston, C., Martin, B., Fichant, G. et al. Bacterial transformation: distribution, shared mechanisms and divergent control. Nat Rev Microbiol 12, 181–196 (2014). The conservation of genes for protection of single strand DNA in *S. aureus* indicates that this is also the case in this species (ref 35).*

*The authors conclude on l. 114 to l 116 that their work “demonstrates that natural competence is conserved in diverse bacterial species beyond fostering genetic variability, specifically to provide bacteria with additional nutritional and metabolic possibilities”. Since the study focused on *S. aureus*, the authors should be careful in extending the conclusion to other bacterial species. While the effector genes are conserved in different species (DNA binding, uptake, etc), there is a general consensus that the environmental cues triggering competence show high variability among competent species. I also think that some points would need to be addressed before arriving to the conclusion on whether this is the case for *S. aureus*. These are specified below, together with other points.*

We agree with this referee that transformation may occur in *S. aureus* and discussed that staphylococcal competence may be associated with the two possibilities, DNA uptake for transformation or as nucleotide source for DNA repair, as these are not two mutually exclusive events (lines 457-459 and 494-498). It is possible that part of the DNA taken up by *S. aureus* cells is degraded and used to repair DNA damage whereas the remaining DNA fragments may recombine with chromosomal DNA. In the case of *S. aureus*, recombination likely occurs, but a number of publications have demonstrated that *S. aureus* is a clonal species and recombination plays a minor role to genetic variation (Fitzgerald Holden 2016, PMID: 27482738; Feil et al. 2003, PMID: 12754228), suggesting that variation from point mutations is 15-fold more frequent than that from recombination. We have clarified these lines in this revised version (lines 523-526).

We also agree with this referee that this study focuses on *S. aureus* have replaced the lines 114-116 according to this reviewer’s suggestion with the following lines: “demonstrates that natural competence is staphylococci goes beyond fostering genetic variability, specifically to provide bacteria with additional nutritional and metabolic possibilities” (lines 116-118). This revised manuscript also includes additional data (Fig. 6), showing the use of exogenous DNA in *S. aureus* cells as a nucleotide source for DNA repair/synthesis.

*(1) While the association between oxidative stress and competence described by the authors is of relevance since this has not been described for *S. aureus* before, this is not unique to *S. aureus*. In *S. mutans*, for instance, oxidative stress has different effects on the competence signaling*

pathway (De Furio M, Ahn SJ, Burne RA, Hagen SJ. Oxidative Stressors Modify the Response of Streptococcus mutans to Its Competence Signal Peptides. Appl Environ Microbiol. 2017 Oct 31;83(22):e01345-17). Not surprisingly, the mechanisms seem to differ, and this is a point that could be addressed in the paper.

As suggested by this referee, we comment on the association that exists between oxidative stress and the induction of competence in other species besides *S. aureus* in this revised manuscript (lines 171-172 and line 178-179). As this referee stated, the growth conditions in which *S. aureus* induces natural competence were unknown; here we show that staphylococcal competence is triggered in response to oxidative stress. While this is an interesting finding, a number of studies described diverse effects of oxidative stress on natural competence in other species. One of our key findings is to show that induction of competence in *S. aureus* increases the glycolytic flux to maintain fermentative growth in conditions of stress and provides a nucleotide source for DNA repair, in addition to the classical role of competence in horizontal gene transfer. This offers a comprehensive view of how natural competence promotes the metabolic flexibility that enables *S. aureus* to colonize the human host.

(2) Growth curves for the different conditions and mutants need to be presented, since some of the differences can be due to collection of data when the bacterial cultures were in different phases, or due to a slow-down in growth. For instance, results on acetate and lactate production can be affected by growth rate, since these represent cumulative values and not real-time production rates. The same is valid for transcriptome data, which is highly influenced by growth phases. Small differences in growth rates can have a large impact on the transcriptome.

According to this referee's recommendation, we included in this revised manuscript additional growth curves of WT, $\Delta comK$ and Kc strains supplemented with different H₂O₂ (supplementary figure S3A). The sampling points over the entire growth cycle are indicated. Furthermore, we refined the description of the acetate/lactate measurements (lines 247 and 301), to clarify that acetate/lactate production levels are represented as a function of culture OD₆₀₀.

In the initial version of this manuscript, we presented data for two strains that showed a lower growth yield than the WT strain; specifically the $\Delta comK$ and NG8 strains. These two strains showed distinct *comK* expression levels and therefore different lactate/acetate levels. The $\Delta comK$ mutant (mutant with no *comK* expression) produced lower levels of lactate/acetate than WT whereas the NG8 strain (induced *comK* expression) produces higher lactate/acetate levels than WT. These observations indicate that a lower growth yield does not condition *comK* expression or the acetate/lactate levels in staphylococcal strains. To reinforce this notion, we have included in the revised manuscript experiments performed using a catalase-deficient strain (*kat* gene) (Supplemental Figure S2A-C and lines 153-158). This mutant is a highly ROS-sensitive mutant and thus shows an increased *comK* expression in regular TSB. In these growth conditions, this mutant showed an induction of *comK* and higher lactate/acetate levels without showing significant growth yield differences compared to the WT. Altogether, these results indicate that there is a direct correlation between lactate/acetate production and *comK* expression, which is not conditioned by growth rate.

(3) Several of the figures show fold change in comK expression. However, comparison with a gene that is not expressed at all, as reported in l. 127 "we were unable to detect comK expression under laboratory growth conditions", should give a fold change of 0.

We have rewritten this sentence according to this referee's suggestion (line 125). *comK* shows a low basal expression level.

(4) Would the authors confirm that the number of samples and replicates was the same for all different experiments, including infection models, macrophage assay, blood/plasma growth,

transcriptome, qPCR, protein interactions, etc, as described in the methods? I suggest this information should be presented for each of the figures.

As suggested by this referee, this revised version of the manuscript shows a standardized number of three biological replicates in almost all experiments (in the initial version of the manuscript, a number of experiments were performed using five biological replicates). In addition, the figures include graphs with individual dots for each biological replicate and the figure legends contain additional information about the number of replicates and the statistical analysis. In general, *in vitro* experiments, such as macrophage assay, blood/plasma growth, transformation assays, oxidant susceptibility assays, bacterial two-hybrid assays, determination of metabolite concentration, the *rif^R* mutation rate, inflammatory cytokines quantification or EA-IRMS show the results of three independent biological replicates. Each biological replicate included three technical replicates. Other *in vitro* experiments, such as immunoblotting, fluorescence microscopy assays, ComK purification or EMSA show a representative results of three biological replicates. The genes with differential gene expression were identified in the transcriptome using two independent datasets and the differential expression of the genes was validated using pPCR analysis by performing three independent biological replicates. Each biological replicate included three technical replicates. *In vivo* infection experiments using *Galleria mellonella* were performed using three independent biological replicates of five larvae. In total, we fifteen larvae were used per tested strain and experiment. For mice infection experiments, we used cohorts of ten mice per tested strain and experiment.

(5) A statistically significant effect in ANOVA is generally followed by additional tests to assess to specific differences. I believe the authors have done it, but this information is not presented in the text.

We present this information in the figure legends of revised manuscript.

(6) The conclusion by the authors that DNA uptake is used for nutrition is not based on direct evidence, but differences in growth in the presence of DNA substrate. As an example, differences in chain formation may affect CFU counts, and pleiotropic effects due to mutations can not be excluded. The fate of DNA needs at least to be investigated in DNA binding and uptake assays, usually with labelled DNA. L.344

We followed this reviewer's recommendation and performed additional experiments to determine the fate of ComG-dependent DNA acquisition (i.e. nucleotide acquisition, carbon acquisition, or repair of DNA damage due to oxidative stress) upon *comK* induction. Specifically, the following experiments are included in this revised manuscript:

1) To determine the fate of exogenous DNA taken up by *S. aureus*, WT, $\Delta comK$, Kc and $\Delta comG$ cultures were supplemented with $^{13}C/^{15}N$ labeled foreign DNA (from *B. subtilis*) and the content of $^{13}C/^{15}N$ stable isotopes was quantified in the chromosomal DNA or the staphylococcal proteins using elemental analyzer isotope ratio mass spectrometry, (EA-IRMS) (Figure 6 and lines 368-386). We detected an important signal in the genome of WT DNA compared to a non-labeled control. The genome of the Kc strain showed higher $^{13}C/^{15}N$ signal than the WT whereas the genome of the $\Delta comK$ or $\Delta comG$ mutants showed a significantly reduced $^{13}C/^{15}N$ signal. In contrast, we did not detect a significant $^{13}C/^{15}N$ signal in the proteome of any strain compared to the negative control.

These results indicate that, in the growth conditions tested, the *comK*-mediated DNA acquisition is not dedicated to the general formation of biomass but is mainly used as nucleotide source for DNA repair/synthesis.

2) To quantify the contribution of DNA taken up by the cells to DNA repair, we quantified the rate of spontaneous mutations acquired in WT and $\Delta comG$ or $\Delta comK$ mutants in the presence of

H₂O₂ (Figure 7C and lines 409-416). The $\Delta comG$ or $\Delta comK$ mutants showed a significantly higher mutation rate than WT. As control, a mutant in the GlcC glucose uptake system show similar mutation rate than WT strain.

Other points:

l. 323 Binding to the comG promoter has been reported before (ref 35)

To our knowledge, there is no report showing staphylococcal ComK promoter binding. Specifically, reference 35 (Fagerlund et al. 2014, PMID: 25155269) uses a transcriptomic approach to identify the genes whose expression is affected by ComK and/or SigH. Whether these genes are regulated in a ComK-direct or indirect manner is not shown in the paper.

l. 379 comk mutant does not survive macrophage assay- but how to know without measuring it?

In the initial version of this manuscript, we showed data regarding the survival of WT, $\Delta comK$ and Kc strains in human monocyte-derived macrophage infection (MDM, MOI 10, figure 1G). We referred to these assays in line 379 as well to provide a general view of all the infection-related data, both *in vitro* and *in vivo* infection experiments. We have refined this statement in line 393 of the revised manuscript.

l. 380 Should not WT and Kc respond in the same way, since ROS should in theory activate the system in the WT?

In general, the response in H₂O₂-treated WT cells is comparable to that of the Kc strain. The Kc is a genetically engineered strain designed to produce a constitutive expression of *comK*, which is a comparable expression to the induction of *comK* detected in H₂O₂-treated WT cells. However, as the Kc strain shows constitutive induction of *comK*, this strain may produce a slightly higher ComK-induced response than the WT strain in certain conditions. We have clarified this in lines 426-427 of this revised manuscript.

l. 385 Blood and plasma should vary in other factors than only blood cells.

To our knowledge, blood centrifugation separates cells (pellet) from plasma (supernatant). Thus, the difference between blood and plasma is that plasma has no cells. Any feedback from this referee on this matter would be greatly appreciated.

-introduce SigH-synergy with comK literature

The synergy of SigH and ComK is now introduced in lines 190-203 of this revised manuscript

-l. 187- rswB deletion mutant had an effect that is presented only in the figure.

RswB is a regulator of SigB activity thus, the effect of *rswB* deletion is attributable to the absence of SigB activity. We clarified this in lines 201 of this revised manuscript.

-L 102 and -l.123- Ref 35 is not correct –DNA transfer and DNA binding were not observed in comK expressing strain

We have reformatted these sentences in this revised manuscript to avoid any misinterpretation.

-The use of anti-oxidants would have strengthened the conclusion that ROS activate the competence response (eg catalase, parakat)

This revised version of the manuscript contains additional experiments of *comK* induction using a catalase-deficient mutant (Supp. Figure S2A-C). As this mutant is more sensitive to ROS, it shows an induction of *comK* expression.

-How do the authors explain the finding that the degree of aeration had no effect on comK expression? L. 134.

During exponential growth of *S. aureus* in TSB cultures, carbohydrates are quite available and *S. aureus* WT cells use fermentation to grow regardless of the degree of aeration. Only when the concentration of carbohydrates decreases, will be bacteria switch to respiration to obtain energy. Such favorable growth conditions to obtain energy do not require additional mechanisms for increasing glucose uptake to feed fermentation, thus *comK* expression under these conditions is repressed. We have clarified this in lines 464-470 of this revised manuscript.

l.l. 96 Reference for L. lactis competence is lacking

We have included a reference (Wydaŭ et al. 2006, PMID: 16553829) for *L. lactis* in this revised manuscript (line 98).

-DNA mutations are also known to trigger competence in S. pneumoniae (Gagne AL, Stevens KE, Cassone M, et al. Competence in Streptococcus pneumoniae is a response to an increasing mutational burden. PLoS One. 2013;8(8):e72613). Authors could mention it.

We thank this referee for this interesting remark, which strengthens some of our statements. We have mentioned this work in this revised manuscript according to the referee's recommendation (lines 407-409).

-Could production of ROS by S. aureus itself contribute to the observed effects?

The amount of ROS produced by the human immune system is orders of magnitude higher than that produced by *S. aureus* through respiration. Thus, it is likely that the ROS produced by the immune system is the mayor contributor of the effects described in this work.

- How does the transcriptome results compare to those presented in ref 35?

There are important differences in growth conditions (Bacto Brain Heart Infusion BHI broth in reference 35 and tryptic soy broth in our assays) that prevented us from making a useful comparison between our results and those presented in reference 35. We are not familiar with the BHI growth medium; whether the concentration of carbohydrates in the BHI medium is high enough to allow fermentation in *S. aureus* cells, which is a key aspect in our study. Knowing specific details of the growth conditions used in reference 35, such as the volume-to-flask ratio, would also be important to make any useful comparison to our study. Nonetheless, the transcriptome of the reference 35 shows induction of a number of glycolytic gene, including the glycine dehydrogenase subunit 1 and 2, (*gcvAP-B*); the glycine cleavage system aminomethyltransferase T (*gcvT*) and the shikimate kinase (*aroK*), in a *comK*-dependent manner.

-l.213- "a large number": please specify it

We have clarified this sentence and refined the description of SNPs localization in this revised version of the manuscript (line 237).

-l 198- introduce comG function

We have introduced the role of ComG in competence in lines 223-224 of this revised version of the manuscript, according to this referee's recommendation.

-Fig 6. % survival results: could it be inhibition of growth, rather than death?

Monitoring mice survival provides an estimation of the inhibition of the progress of the infection. The immune system of mice prevents the infection to progress, probably causing both inhibition of growth and bacterial death.

-Fig 6 (B) Was there growth or death in blood and plasma? Usually in these assays a mutant that can not divide in blood is used, to exclude growth as a confounding factor.

The possibility that bacteria grow in this assay is unlikely. For this assay, we used the classical protocol in which 10^6 CFU are resuspended in PBS buffer containing 5% human blood or plasma (Morrison et al. 2012, PMID: 22493015; Yepes et al. 2014, PMID: 24747904). Samples were incubated for 25 min before serial dilution and plating. In principle, the amount of nutrients and incubation period does not allow staphylococcal growth. In addition, *S. aureus* cells are highly stressed during incubation with blood and cell division is blocked in these incubation conditions (Malachowa et al. 2011, PMID: 21525981; Painter et al. 2017, PMID: 28993457). Thus, we recovered a CFU count much lower than the inoculum, pointing to bacterial death as a critical factor of the phenotypic differences observed in this assay. We have clarified this point in lines 761-765 of this revised manuscript.

-NG8 strain: it would help the reader if the strain background is explained in the figure legend

We have included the details of the NG8 strain background in the figure 2 legend of the revised manuscript, according to this referee's suggestion.

-Could the NG8 phenotype showing changes in acetate, lactate and O₂ consumption be a result of an altered growth rate? I miss a growth curve for NG8, and also the other strains, particularly in the presence of H₂O₂.

Growth curve of NG8 and WT strain was shown in figure 2D of the initial version of the manuscript (lines 229-231). In the revised manuscript, we have included growth curves of WT, $\Delta comK$ and Kc strains with different H₂O₂ concentrations (Supplemental Figure S3A).

As we explained in a previous point of this response letter (please see above answer to point 2, referee 2), the growth yields do not condition the acetate/lactate production in our assays. The $\Delta comK$ and NG8 strains both showed lower growth yields than the WT and yet, these two strains showed distinct *comK* expression and different lactate/acetate levels. The $\Delta comK$ mutant showed no *comK* expression and produced lower levels of lactate/acetate whereas the NG8 strain showed induced *comK* and produced higher lactate/acetate levels. To strengthen this notion, this revised manuscript contains a supplemental figure using a catalase-deficient strain (*kat* gene) (Supplemental Figure S2A-C). This mutant is a ROS-sensitive mutant; it shows an induced *comK* expression and higher lactate/acetate levels without showing significant differences in growth rate compared to the WT. Overall, there is a direct correlation between lactate/acetate production and *comK* expression that is not conditioned by differences in growth rate.

-l.254- a Venn diagram showing the common genes increased in the Kc vs WT and reduced in the comK deletion vs WT would be helpful.

The revised manuscript includes a comparison of the genes with increased expression in the Kc vs WT or reduced expression in the $\Delta comK$ vs WT, particularly those involved in glycolysis, pentose phosphate pathway, metabolism of purines and pyrimidines or TCA (supplementary figure S8A). *comK* induction leads to an increase in the expression of genes related to carbohydrate metabolism whereas TCA genes remain uninduced.

-It would be a valuable information to know whether comK is expressed in vivo. Have the authors measured it?

We have not measured *comK* expression during *in vivo* infections. However, our *in vitro* infection data shows that *comK* expression is induced during infection of human alveolar epithelial cells (Fig 1G), thus it demonstrates the expression of *comK* in conditions that would be encountered during *in vivo* infections. In addition, we demonstrate the *in vivo* relevance of *comK* using *in vivo* infection of mice and *Galleria mellonella* (Fig. 8C-D and supplemental figure S12A-B).

-Fig. 7. The information in the legend does not match with the figures. Looks like A and B is inverted.

We have modified the figure legend accordingly in this revised manuscript.

-Fig. 3. L 1066-1067: looks like the information in the legend is inverted.

We have corrected the figure legend in this revised version of the manuscript.

-fig 3 A and B, the comparison description in the figures is different from the description in the legend (WT vs delta, delta vs WT)

We have rewritten this sentence in the figure legend according to this reviewer's suggestion.

Reviewer #3 (Remarks to the Author):

The function of genes in many gram positive species, including S. aureus, that are orthologous to those related to the induction of competence and transformation in Bacillus subtilis and S. pneumoniae are not well established. The investigative team has found that the competence pathway in S. aureus is induced (via ComK) under oxidative stress and that a comK allelic replacement mutant is less virulent in both murine and Galleria infection models. This is an original and impactful observation. Previous literature has documented the ability of S. aureus to facilitate DNA transformation upon induction of SigH (which was comG dependent).

Typically comK is not induced during growth in rich medium such as TSB. However, upon induction of comK, S. aureus grows via a fermentative mechanism. Appropriate experiments were performed to document this (acetate, lactate, NAD/NADH ratio, oxygen consumption). Further, it was determined that induction of comK increases glucose consumption, it is probable that this is via direct binding of ComK and subsequent induction of genes encoding glucose transporters. Thus, the evolving model proposed by the investigators is that upon oxidative stress, comK is induced and one outcome is that glucose transporters are induced further increasing glycolysis as respiration is inhibited (presumably via the oxidative stress). As such, comK is required for infection in both murine and Galleria models.

However, although this is novel and impactful, the investigators did not necessarily define how the competence network and DNA uptake is important for the phenotypes described above. Clearly, this is the most impactful question as glucose catabolism is known to be very important for initial S. aureus infection. It seems that the investigators need to determine if upon comK induction whether ComG-dependent DNA acquisition are required for nucleotide acquisition, carbon acquisition, or required to fix DNA damage due to oxidative stress. Experimental approaches to address these questions are not necessarily difficult to address (13C/15N experiments or transformation experiments in defined media lacking nucleotides). Further, to delineate if nucleotide acquisition is required for the phenotypes associated with the animal models of infections one could utilize the comG mutant with or without comK mutation. Since ComK function to induce glucose acquisition, the lack of virulence would be predicted.

We have followed this referee's recommendation and have included additional experiments showing the importance of DNA uptake for the infection-related phenotypes described in this work. Specifically:

1) To determine the fate of the exogenous DNA taken up by *S. aureus* cells, we grew WT, $\Delta comK$, Kc and $\Delta comG$ strains and cultures were supplemented with $^{13}C/^{15}N$ labeled foreign DNA (from *B. subtilis*). The content of $^{13}C/^{15}N$ stable isotopes was quantified in the chromosomal DNA or the staphylococcal proteins using elemental analyzer isotope ratio mass spectrometry (EA-IRMS) (Figure 6 and lines 368-386). We detected an important signal in the genome of WT DNA compared to a non-labeled control. The genome of the Kc strain showed higher $^{13}C/^{15}N$ signal than the WT whereas the genome of the $\Delta comK$ or $\Delta comG$ mutants showed a significantly reduced $^{13}C/^{15}N$ signal. These results indicate that *comK* induces a *comG*-dependent acquisition of DNA as nucleotide source to repair DNA damage. In contrast, we did not detect a significant $^{13}C/^{15}N$ signal in the proteome of any strain compared to the negative control, suggesting that DNA acquisition is not directed to general biomass formation but mainly used as nucleotide source in the growth conditions tested.

2) To determine the accumulation of DNA damage in the different *S. aureus* genetic backgrounds, we quantified the rate of spontaneous mutations acquired in WT and mutants defective in *comG*, *comK* or *glcC* in the presence of H_2O_2 (Figure 7C and lines 409-416). Cultures were serially diluted and plated in TSB supplemented with rifampicin to identify spontaneous mutations in *rpoS* gene that confer resistance to rifampicin. $\Delta comG$ or $\Delta comK$ mutant showed a significantly higher mutation rate than WT or the mutant defective in *GlcC* glucose uptake system. This result indicates that *ComG*-dependent DNA uptake is required to repair the DNA damage that is caused by oxidative stress.

3) We have used an invertebrate animal infection model to compare the virulence potential of WT, $\Delta comG$, $\Delta comK$ and the $\Delta comG \Delta comK$ double mutant (Supplemental Figure S12Ai). The *comK*-defective mutants ($\Delta comK$ and the $\Delta comG \Delta comK$) showed a significant attenuation of virulence. The $\Delta comG$ mutant also showed an attenuation of the virulence. These results are consistent with our hypothesis that *comK* regulates glucose and DNA uptake during infection.

Below are some specific comments.

1. Previous manuscripts have documented that the competence system is induced via SigH. This aspect was not necessarily addressed in this manuscript. Is SigH required for any of the phenotypes discussed in the manuscript? This is important to note one way or the other.

In this revised manuscript, we have included additional results that show the interplay between SigH and ComK (Fig 1Hi and Supplemental Figure S6A, lines 190-203). The Supplemental Figure S6A shows no *comK* induction in the $\Delta sigH$ mutant. Therefore, this revised manuscript includes additional experiments to compare the survival (CFU count) of WT, $\Delta sigH$, $\Delta comK$ or $\Delta sigH \Delta comK$ strains in the presence of H_2O_2 (Supplemental Figure S6B). All mutants showed a growth defect in comparison to WT, suggesting that *sigH* contributes to *comK* induction and thus, to the phenotype that we presented in this work. Accordingly, we tested the virulence potential of these strains using an invertebrate animal model. Results show that all mutants are strongly impaired in *in vivo* infection (Supplemental Figure S12Aii). Results are described in lines 443-444 of this revised manuscript.

2. Line 459 and 460. The authors state that they documented that DNA is used for a nutrient source? Not sure this has been documented nor have the authors documented whether DNA can act as a carbon source (or even a nitrogen source) in S. aureus.

The role of DNA as a nutrient source was initially proposed by RJ Redfield (1993) and followed by a number of publications from different laboratories (Finkel and Kolter, 2001; Palchevskiy and Finkel, 2006 and Chimileski et al., 2014). We have clarified this statement in this revised manuscript (lines 90-94 and 320-322). This revised version of the manuscript also includes additional experiments using stable-isotope DNA labeling (Fig. 6) and analysis of spontaneous DNA mutations in various *S. aureus* mutants (Fig. 7C), showing that part of the DNA taken up during natural competence is used as nucleotide source for DNA repair/synthesis (as described above).

3. In the materials and methods section, the transformation experiments were not well described. What was the plasmid used? Where was lacZ inserted in the chromosome and what lacZ gene was utilized. Further, Morikawa and colleagues utilized 10 micrograms of chromosomal DNA (typically has to be spooled to generated this concentration). This may be why the authors could not detect transformants using chromosomal DNA.

We expanded the description of the transformation conditions in the Methods section (lines 585-593) according to the reviewer's suggestion. Briefly, transformation with plasmids involves the use of the pAmy plasmid (Yepes et al. 2012, PMID: 24747904); a derivative of pMAD plasmid (Arnaud et al. 2004, PMID: 15528558) that is suitable for chromosomal recombination. The *lacZ* gene that we used is that of the pMAD plasmid. For transformation with genomic DNA, we used chromosomal DNA from an engineered *S. aureus* strain, in which the *lacZ* and *erm* genes from pMAD were integrated in the *amyE* locus (SA2244) by double recombination.

We used 2 µg of chromosomal DNA in the transformation experiments presented in this work. We agree with the reviewer's remark that increasing the concentration of chromosomal DNA would increase the number of transformants. However, it is not our intention to optimize the process of staphylococcal transformation but to emphasize that recombination is a challenging event, possibly because part of the DNA taken up by the cells is degraded and used for a different purpose. With the experiments presented in this work, we show that part of the DNA taken up by the cells is used as nucleotide source for DNA repair/synthesis. We have clarified this point in the Discussion section of this revised manuscript (lines 473-476 and lines 498-500).

4. Lastly, if chromosomal DNA is acquired and used as a source to repair DNA, one would expect to detect this process using population biology techniques. These studies to date have found that S. aureus does not evolve through recombination. This should be discussed in the discussion section.

We thank this referee for this interesting remark, which strengthens some of our statements. We have included this point in the Discussion section of this revised manuscript (lines 523-26).

Reviewers' Comments:

Reviewer #1:

Remarks to the Author:

The authors have adequately addressed all of my previous concerns.

Reviewer #2:

Remarks to the Author:

The authors have addressed well the raised questions. This is a study that brings important advances in the field.

Reviewer #3:

Remarks to the Author:

No further comments, all comments have been appropriately addressed. Bravo!